# Light microscopy based approach for mapping connectivity with molecular specificity

Fred Y. Shen [1,2], Margaret M. Harrington[3], Logan A. Walker [4], Hon Pong Jimmy Cheng[3], Edward S. Boyden [5,6] & Dawen Cai [2,3,4 ✉]

Mapping neuroanatomy is a foundational goal towards understanding brain function. Electron microscopy (EM) has been the gold standard for connectivity analysis because nanoscale resolution is necessary to unambiguously resolve synapses. However, molecular information that specifies cell types is often lost in EM reconstructions. To address this, we devise a light microscopy approach for connectivity analysis of defined cell types called spectral connectomics. We combine multicolor labeling (Brainbow) of neurons with multi-round immunostaining Expansion Microscopy (miriEx) to simultaneously interrogate morphology, molecular markers, and connectivity in the same brain section. We apply this strategy to directly link inhibitory neuron cell types with their morphologies. Furthermore, we show that correlative Brainbow and endogenous synaptic machinery immunostaining can define putative synaptic connections between neurons, as well as map putative inhibitory and excitatory inputs. We envision that spectral connectomics can be applied routinely in neurobiology labs to gain insights into normal and pathophysiological neuroanatomy.

[1] Medical Scientist Training Program, University of Michigan, Ann Arbor, MI, USA. [2] Neuroscience Graduate Program, University of Michigan, Ann Arbor, MI, USA. [3] Department of Cell and Developmental Biology, University of Michigan, Ann Arbor, MI, USA. [4] LS & A, Program in Biophysics, University of Michigan, Ann Arbor, MI, USA. [5] McGovern Institute, Koch Institute, Department of Media Arts and Sciences, Department of Biological Engineering, and Department of Brain and Cognitive Sciences, Massachusetts Institute of Technology, Cambridge, MA, USA. [6] Howard Hughes Medical Institute, Cambridge, MA, USA. ✉email: dwcai@umich.edu

Mammalian brains are extraordinarily complex pieces of circuitry, composed of trillions of connections between diverse cell types. Understanding these wiring patterns is a fundamental piece of the puzzle toward understanding how our brains work. A comprehensive wiring diagram of neuronal connections is called a connectome, while the pursuit of a connectome is known as connectomics. Neurons are connected to each other primarily through chemical synapses. They are located where presynaptic and postsynaptic neurons touch, and are composed of structures spanning hundreds of nanometers in size. The presynaptic compartment contains a region that mediates neurotransmitter release called the active zone[1]. The postsynaptic compartment contains a region that is protein dense called the postsynaptic density (PSD), which is comprised of neurotransmitter receptors, scaffolding proteins, and signaling molecules[2].

Due to their small size, neuroscientists have relied on electron microscopy (EM) to observe chemical synapses and map connections between neurons. EM is a powerful tool that offers unparalleled nanometer resolution[3], sufficient to observe synaptic structures. Recent works have used EM to define the adult Drosophila hemibrain connectome[4], as well as curate a $92.6 \times 94.8 \times 61.8\ \mu m^3$ connectome from mouse somatosensory cortex[5]. Despite the recent technological and biological advances with EM, several challenges remain. Molecular information is lost as proteins can only be rarely identified with EM alone. Long imaging timespans generate large datasets that require demanding computational processing and analysis. All of these challenges have placed the use of EM for connectomics beyond the reach of common neurobiology labs. Moreover, because of the aforementioned difficulties in scaling the technology, the use of EM for connectomics is not currently suitable for high throughput experimentation. Given that neuroanatomy can vary between animals and substantially change in disease states, a new approach is needed for connectivity analysis that considers molecular information and is easily scalable.

In contrast to EM, light microscopy (LM) can generate specific molecular information through immunostaining, but lacks the nanoscale resolution to map synapses. Super-resolution light microscopy (srLM) techniques, such as STORM[6–8] or STED[9], possess nanoscale resolution and molecular specificity, but are not suitable for thick brain volumes because of slow imaging speeds, photobleaching, and optical distortions. The recent development of expansion microscopy[10–12], which grants super-resolution to routine LM imaging through isotropic, physical magnification of hydrogel-tissue hybrids, offers an alternative path forward that can combine molecular specificity, nanoscale resolution, and rapid LM in thick brain volumes[13].

Here, we describe a strategy to obtain high throughput morphology measurements of densely labeled neuronal populations, with integrated molecular and connectivity information for multimodal analysis. We combine a multicolor genetic labeling tool (Brainbow) with a multi-round immunostaining Expansion microscopy (miriEx) strategy to simultaneously profile single neuron morphologies, molecular marker expression, and connectivity in the same brain section. We define the derivation of these properties from hyperspectral fluorescent channels as spectral connectomics, a LM based approach towards mapping neuroanatomy and connectivity with molecular specificity.

## Results

### Spectral connectomics encodes multimodal neural information. Spectral connectomics is based on the ability to acquire multichannel LM datasets at nanoscale resolution that encode neuron morphologies, cell-type profiles, and connectivity. Thus,

we needed to develop a strategy for (1) dense labeling of a neuronal population with the ability to unambiguously trace dendrites and axons, (2) multiplexed readout of important protein markers, and (3) multiscale imaging that can range from nanoscale resolution for resolving synapses, to microscale resolution for resolving cell-type markers. To do so, we first optimized an expansion microscopy protocol for multi-round immunostaining (miriEx) that let us generate multichannel LM datasets at multiple resolutions. We then added Brainbow to stochastically express fluorescent proteins (FPs) in a cell-type-specific population of neurons[14,15]. The combination of miriEx and Brainbow allowed us to curate hyperspectral, multiscale LM datasets that contain information about molecular markers, neuron morphologies, and synaptic machinery (Fig. 1a).

### miriEx—multiplex immunostaining for spectral connectomics. We start by describing the development of miriEx for multiplexed immunostaining. Probing multiple proteins using traditional immunohistochemistry (IHC) is typically limited by host animal species of primary antibodies (most tend to be either mouse or rabbit) and visible light bandwidth, such that detecting more than four targets becomes difficult. Recent strategies, such as Immuno-SABER[16], PRISM[17], and CODEX[18] have been developed to overcome these limitations using antibody-DNA barcoding and readout strategies. Alternatively, tissue sections can undergo multiple rounds of routine antibody staining, imaging, stripping, and restaining, as seen in array tomography[19], CLARITY[20], MAP[21], SWITCH[22], and SHIELD[23]. We adopted the latter strategy and optimized a protein crosslinking protocol to anchor antigens into an expandable hydrogel. Specifically, we used acrylic acid N-hydroxysuccinimide ester to modify proteins with acryl groups so they can be crosslinked and polymerized into an expandable hydrogel. We replaced the Proteinase K digestion step from standard expansion microscopy protocols[10,12] with SDS/heat based denaturation because Proteinase K destroys endogenous proteins. In contrast, SDS/heat treatment preserves endogenous proteins and is compatible with post-gelation immunostaining[21]. Furthermore, because our denaturation method is similar to that of SDS-PAGE western blots, we found that antibodies already validated for western blots usually work with miriEx (Supplementary Table 2). We also found that the same SDS/heat treatment can efficiently strip antibodies after each round of probing (Supplementary Fig. 1). We validated that gel-tissue hybrids in miriEx expand ~2× in 1× PBS and ~4× in 0.001× PBS respectively (Supplementary Fig. 2).

We then demonstrated that miriEx robustly preserves antigens across multiple rounds of immunostaining and stripping. Conceptually, different rounds of staining can be used to probe different neuronal properties, such as molecular markers, morphology, and/or synaptic markers (Fig. 1a). We probed five different rabbit antibodies across 7 rounds of imaging in one basolateral amygdala (BLA) sample (Fig. 1b), imaging DAPI in each round to use as a fiduciary channel for registration (Supplementary Fig. 3). We quantified the signal to noise (SnR) of PV immunostaining in rounds 1, 4, and 7 and found that there was a larger decrease from round 1 to round 4 than from round 4 to round 7 (Fig. 1c). This suggests that the amount of protein lost in the antibody elution step decreases after every round. The fact that round 7 PV SnR is still one third of round 1 PV SnR even after 6 rounds of elution indicates that antigens are still preserved in later rounds of miriEx and can be retrieved. To extend the application of multi-round immunostaining, we showed that miriEx works with formalin-fixed human tissue (Supplementary Fig. 4). We also demonstrated that through multispectral imaging for 2–3 antigens in each round, we could achieve highly

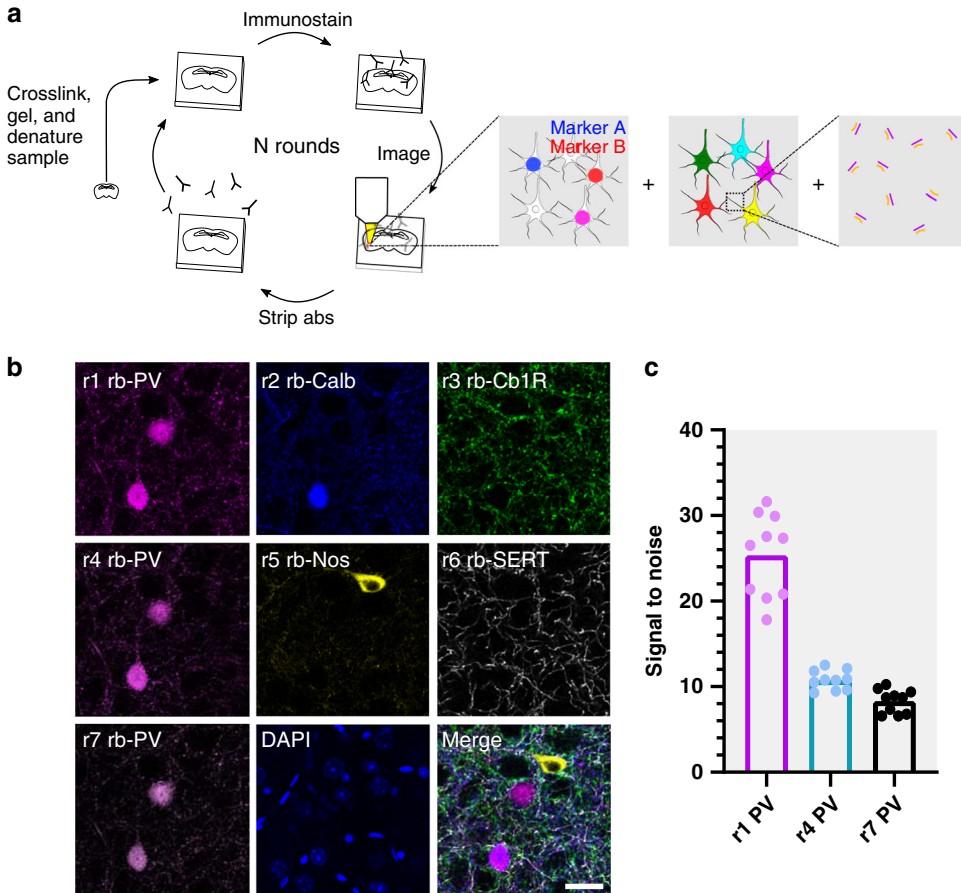

**Fig. 1 miriEx robustly preserves antigens across multiple rounds of immunostaining, imaging, and stripping. a** miriEx strategy to simultaneously measure molecular profile, morphology, and/or connectivity. Tissue samples are embedded in a hydrogel to create expandable gel-tissue hybrids. The sample can then undergo multiple rounds of immunostaining, imaging, and stripping to measure various neuron properties. We then register and merge the different rounds of imaging to correlate the results. **b** Five distinct rabbit antibodies were used across 7 sequential rounds of immunostaining, with PV re-probed in rounds 4 and 7 to demonstrate retention of antigen. The merged image shows r1 PV, r2 Calb, r3 Cb1R, r5 NOS, and r6 SERT. **c** Signal to noise was quantified for PV immunostaining in round 1, round 4, and round 7 ($n = 10$ neurons). Abs antibodies, PV parvalbumin, Calb calbindin, Cb1R cannabinoid receptor type 1, Nos nitric oxide synthase, SERT serotonin transporter. Scale bar: (**b**) 25 μm (pre-expansion size). Expansion factor: (**b**) ~2×. See Supplementary Table 1 for more details.

multiplexed profiling of 15 different targets in the same piece of mouse striatum tissue (Supplementary Fig. 5).

In the BLA multi-round immunostaining experiment described above, we observed that although the majority of amygdala PV neurons co-expressed calbindin (Calb), some were calbindin negative (Fig. 1b). Past studies indicate that these PV+/Calb− neurons are axo-axonic cells that specifically innervate the axon initial segment[24], while PV+/Calb+ neurons represent basket cells that innervate the perisomatic region[25]. Somatostatin (SOM) also marks another broad interneuron subtype in the amygdala, with both Calb positive and negative co-expression[26]. As a result, we chose this system to demonstrate the ability to interface molecular marker information with morphology analysis using miriEx.

**Profiling neuron morphology with molecular specificity.** To differentiate intermingled neurons in situ, we used Brainbow, a technique that relies on the stochastic expression of FPs to label neighboring neurons in unique colors[14,15]. Importantly, (1) the FPs used are distinct antigens, allowing their signal to be amplified through miriEx immunostaining, (2) the FPs are membrane targeted, which enables better labeling of neuron subcellular morphology[15], and (3) dense labeling of a neuronal

population can be achieved to study morphology in a more high throughput manner compared to other techniques that rely on sparse labeling. *PV*-Cre and *SOM*-Cre double transgenic animals were generated to allow genetic access to two broad interneuron types. Brainbow AAVs 2/9 were stereotaxically injected in the BLA, and 200 μm sections of tissue were processed with miriEx (Fig. 2a). In round 1, three different molecular markers (Calb, PV, and SOM) were probed to define four molecular cell types: PV, PV/Calb, SOM, and SOM/Calb (Fig. 2b–e). In round 2, three Brainbow FPs were immunostained to reveal morphology (Fig. 2f–i). DAPI was co-stained as a fiduciary channel for registration of the two rounds. Both rounds of imaging took place with the gel-tissue hybrid expanded ~2× in 1× PBS, giving us an effective imaging resolution of ~150 × 150 × 350 nm³. Brainbow AAVs labeled 2 out of 4 (50%) PV neurons, 20 out of 31 (70%) PV/Calb neurons, 7 out of 7 (100%) SOM neurons, and 24 out of 28 (86%) SOM/Calb neurons within a 590 × 404 × 160 μm³ volume. Following identification of each Brainbow neuron by its molecular subtype, we reconstructed its dendritic morphology using nTracer[27], an ImageJ/Fiji plugin for tracing multispectral datasets (Fig. 2j, k, Supplementary Movie 1, manual and tutorial videos can be found at https://www.cai-lab.org/ntracer-tutorial). A total of 53 neurons in the imaged volume was reconstructed across all four molecular subtypes and various morphology

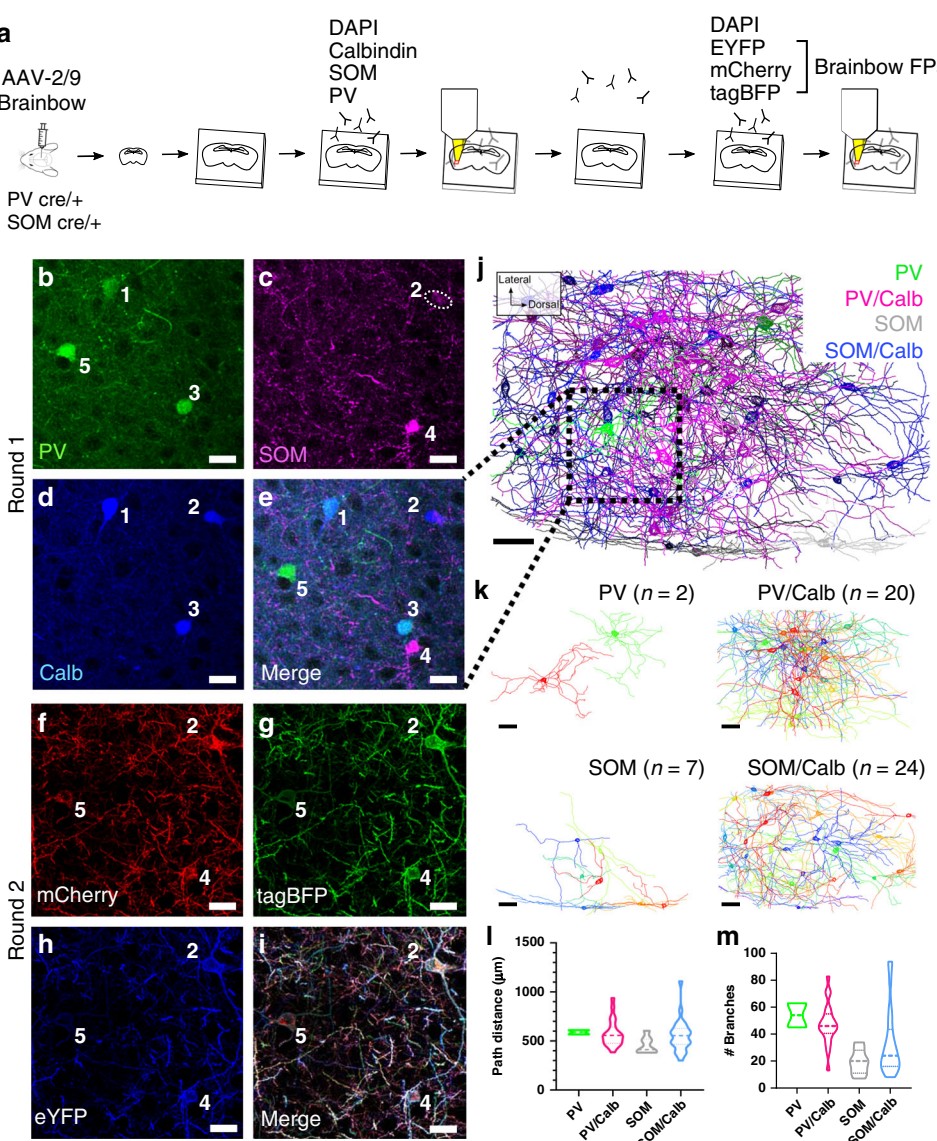

**Fig. 2 miriEx combined with Brainbow to simultaneously profile molecular expression and neuron morphologies. a** Experiment design: Brainbow and molecular markers were imaged across two rounds of immunostaining using the DAPI channel for registration. **b–e** MIP showing the molecular markers (PV, SOM, Calb) imaged in round 1. Four distinct molecular subtypes could be identified: PV, PV/Calb, SOM, SOM/Calb. (**f–i**) MIP showing the Brainbow channels imaged in round 2. **j** nTracer reconstruction of 53 neurons encompassing the four subtypes previously identified in a 590 × 404 × 160 $\mu m^3$ volume. The red square represents the field of view seen in (**b–i**). **k** Individual nTracer reconstructions based on molecular subtype. **l** Total path distance plotted for each of the molecular subtypes. **m** Number of dendritic branches plotted for each of the molecular subtypes. Violin-plot: bold dashed line, median; dashed line, upper and lower quartiles. MIP maximum intensity projection, PV parvalbumin, Calb calbindin, SOM somatostatin. Scale bars: (**b–i**) 25 μm (pre-expansion size). (**j**, **k**) 50 μm (pre-expansion size). Expansion factor: (**b–i**) ~2×. See Supplementary Table 1 for more details.

parameters were analyzed (Fig. 2l, m, Supplementary Fig. 6). In agreement with past findings, PV expressing neurons appear to have more complex branching patterns compared with SOM expressing neurons, despite having similar dendritic lengths[25,26]. This type of reconstruction enables study of how different neuronal cell types interact with each other anatomically, making it a powerful tool for morphology analysis. For a technical replicate, we repeated a similar experiment to reveal PV, PV/Calb, SOM, and SOM/Calb dendritic morphology in the dorsal endopiriform nucleus (Supplementary Fig. 7).

**Gephyrin–Brainbow–Bassoon trio defines inhibitory synapses.** While understanding the projection patterns of axons and dendrites is an important aspect of mapping neuroanatomy, another

central objective is to understand how neurons connect. Chemical synapses are hundreds of nanometers in size and serve as bridges of communication between neurons[7]. Measuring synapses using conventional LM techniques is difficult as the distance between synaptic structures and neuronal boundaries can be smaller than the diffraction limit[6]. Recently, expansion microscopy has been shown to be a viable strategy for resolving synaptic structures and assigning them to neurons[13,28,29]. Consequently, we combined Brainbow with miriEx to measure synaptic structures located at the junctions between different neurons in an effort to define connectivity using LM.

We first confirmed that miriEx was compatible with endogenous synaptic machinery immunostaining by probing for Gephyrin (inhibitory PSD), Homer1 (excitatory PSD), and Bassoon (presynaptic active zone) in layer 4 somatosensory

cortex (Supplementary Fig. 8). ~4× expansion of the sample gave us an effective imaging resolution of $\sim 70 \times 70 \times 200\ nm^3$ using confocal microscopy. We found inhibitory Gephyrin–Bassoon synapse pairs to be less common (21%) than excitatory Homer1–Bassoon synapse pairs (77%). Importantly, only ~2% of the Bassoon puncta were not paired with Gephyrin or Homer1, which were mutually exclusive (Supplementary Fig. 8). This gave us confidence that the Gephyrin and Homer1 antibodies we used mark inhibitory and excitatory synapses almost completely. We also showed that over 90% of inhibitory and excitatory synapses are >300 nm away from their closest neighbor, allowing us to reliably distinguish neighboring synapses at our imaging resolution (Supplementary Fig. 8).

We then packaged Brainbow in AAV-PHP.eB[30] serotype to efficiently transduce neurons systemically across the brain via intravenous injection. We retro-orbitally injected AAV-PHP.eB Brainbow in *PV*-Cre mice and found that we had near complete coverage of PV neurons in somatosensory cortex (Supplementary Fig. 9). 100% of the Brainbow labeled neurons were positive for PV immunostaining indicating that our labeling strategy was both highly sensitive and specific (Supplementary Fig. 9). 100 μm sections of somatosensory cortex were processed with miriEx. In round 1, three Brainbow FPs were stained and the sample was expanded ~4× and layer 4 was imaged (Fig. 3a, b). In round 2, a presynaptic marker (Bassoon), inhibitory postsynaptic marker (Gephyrin), and EYFP were stained, and the sample was again expanded ~4× and imaged in layer 4 (Fig. 3c). We observed that Bassoon–Gephyrin pairs could be resolved and were located at axosomatic and axodendritic contact points between Brainbow labeled PV neurons (Fig. 3d–i, Supplementary Movies 2, 3). Measuring the line profile of these putative synapses revealed that Gephyrin, the postsynaptic Brainbow membrane, and Bassoon were arranged in the expected order (Fig. 3p, q). The distance between Gephyrin and Bassoon puncta was between 100 and 200 nm and matched previous reports[7,28]. Historically, ultrastructural features from EM images (i.e., synaptic vesicles, postsynaptic density, and synaptic cleft) have been used to define synapses. Recent advances in super-resolution LM have demonstrated that pre- and postsynaptic proteins themselves can provide an alternative definition of a synapse[13,19]. Assuming the average distance between Gephyrin and Bassoon puncta is ~150 nm[7,28], 4× expansion enables confocal imaging resolutions of $75 \times 75 \times 200\ nm^3$ which sufficiently meets Nyquist sampling to resolve Gephyrin–Bassoon pairs in the lateral directions, but not completely in the axial direction. The synaptic cleft between pre- and postsynaptic Brainbow membranes is tens of nanometers wide, however, was not resolvable at this resolution. But our work pushed further to use the trio of Gephyrin–Brainbow–Bassoon signals, similar to EM does, to give us confidence to define putative inhibitory synapses between Brainbow labeled PV neurons.

In the ideal case, synapses identified with spectral connectomics would be directly verified by correlated EM imaging. However, it is beyond the scope of this paper as EM is not compatible with expansion protocol due to membrane extraction. That said, if multiple putative synapses are identified between the exact two neurons, from a statistics perspective, it is more likely that these two are functionally connected. We believe it is very helpful to identify potential connections between two neurons unequivocally. Moreover, our approach towards spectral connectomics allowed us to eliminate false positives that would not be possible with diffraction-limited LM. Neuronal contacts have previously been used as correlates for synaptic sites[27,31], but we show an example of a close apposition lacking synaptic machinery that would be falsely classified as a connection (Fig. 3j–l, r). Another type of error that could arise is the wrongful

assignment of a physically close axon that merely passes through and does not form a synapse (Fig. 3m–o, s). Given that axons can be small caliber with bouton sizes that are hundreds of nanometers[32], diffraction-limited LM may confuse neighboring axons, leading to synapse assignment mistakes.

Following the first two rounds of miriEx that probed Brainbow and synaptic markers, a third round of miriEx was performed to probe SOM, adding information about molecular cell types to the dataset (Supplementary Fig. 10). By doing so, we show the ability to map PV connectivity onto a variety of cell types. However, it remains difficult to accurately assign synapses without both the presynaptic and postsynaptic Brainbow membranes. Without a postsynaptic SOM neuron membrane label, it is challenging to tell if the PV axon synapses directly on the soma or on an unlabeled, small caliber dendrite sandwiched in between. As a result, we did not attempt to analyze PV to SOM connectivity.

**Connectomic analysis of a molecularly defined cell type**. After validating that putative inhibitory synapses could be identified between Brainbow labeled PV neurons, we set out to trace the axons and dendrites of 8 PV neurons whose somas were located inside a $\sim 100 \times 100 \times 60\ \mu m^3$ imaging volume (Fig. 4a). We then traced every Brainbow labeled PV axon that innervated these eight PV neurons, and annotated all the putative inhibitory synapses (Gephyrin–Brainbow–Bassoon trios) that we could identify. One hundred eighty nine axons were traced and 422 molecularly specific (PV–PV) putative synapses were defined (Fig. 4b, Supplementary Fig. 11, Supplementary Movie 4). First we analyzed the connections between these 8 PV neurons by plotting their connectivity matrix (Fig. 4c). We observed that neuron 37 innervated 4 other PV neurons, matching previous reports of local PV–PV connectivity[33]. Interestingly, while local PV–PV connection is common in our dataset, we did not find local reciprocal inhibition between two PV neurons. We did observe an example of indirect connectivity where neuron 37 connects with neuron 1 indirectly through neuron 6, highlighting that neuron assemblies can be mapped with spectral connectomics. Next we added 189 traced PV axons, whose somas were not located in this volume, to the same connectivity matrix (Fig. 4d). We found multiple examples of presynaptic PV axons that innervate more than one postsynaptic PV neuron. However, many of the dimly labeled PV axons could only be reliably traced for a short distance, leaving the full extent of their synaptic connections uncovered. This likely skewed the connectivity matrix towards under-representing PV axons that synapse onto multiple postsynaptic PV neurons. To demonstrate reproducibility of the technique, we repeated the same experiment on another sample and reconstructed 7 PV neurons, 223 PV axons, and 332 PV–PV synapses (Supplementary Fig. 12).

Compared to traditional monochromic labeling, Brainbow labeling provides spectral information to identify the source of innervating axons. For example, we can be confident that two spectrally unique axons come from different PV neurons even without tracing them back to their somas. Same colored axons are more difficult to interpret as they could come from different neurons that happen to have similar colors, or they could originate from the same neuron branch outside our imaging volume. Nevertheless, we can still use color to estimate the upper and lower bounds when asking the convergence question of how many unique PV neurons innervate one PV neuron. The upper bound is determined by the number of innervating axons, while the lower bound is determined by the number of unique colors these axons possess (Fig. 4e). To determine the number of unique colors, we plotted their RGB color on a ternary plot and used an elbow plot to conservatively estimate the number of k-means

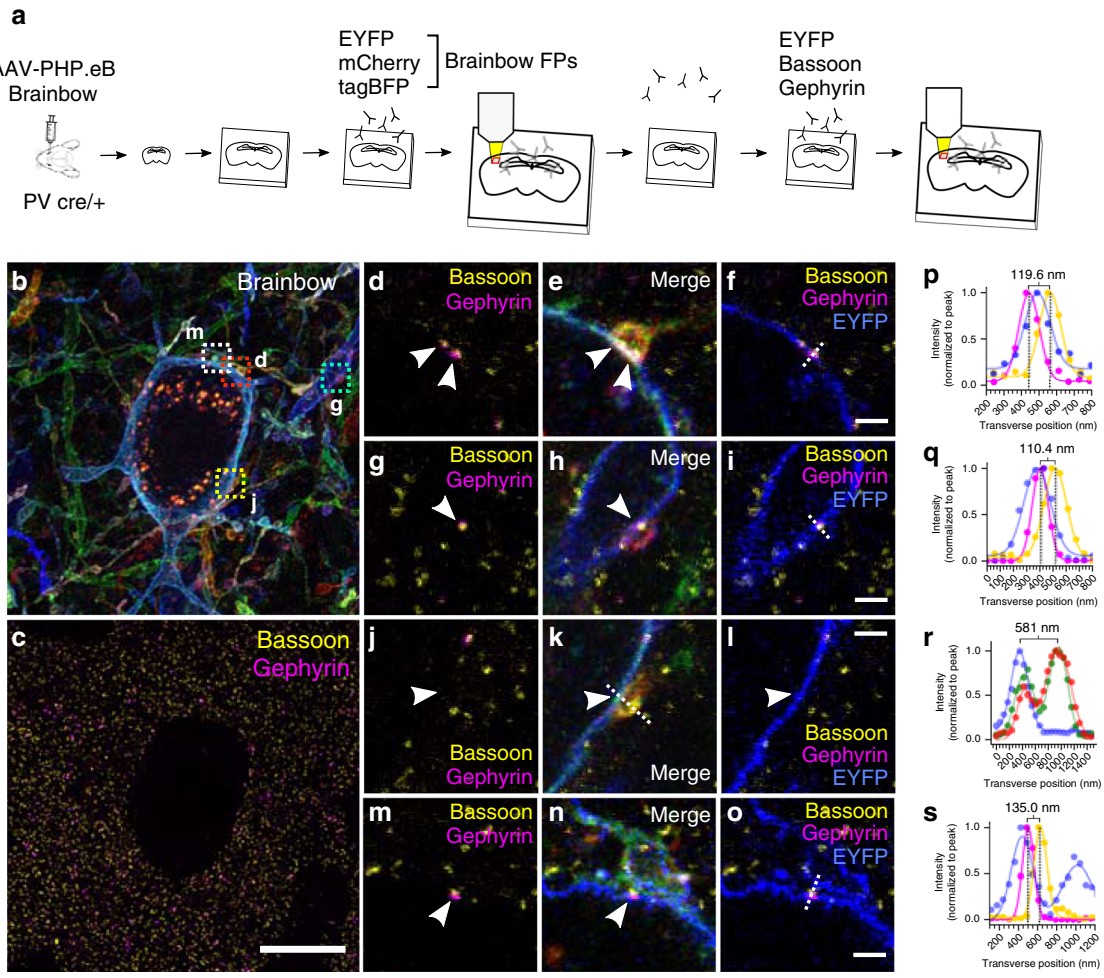

**Fig. 3 Putative synaptic connections can be defined between Brainbow neurons. a** Experimental design: Brainbow FPs and endogenous synaptic markers are imaged across two rounds of immunostaining using the EYFP channel for registration. **b** MIP showing the Brainbow channels imaged in round 1. **c** MIP showing the Bassoon and Gephyrin puncta imaged in round 2. **d**, **f** Single slice zoomed inset of red square in (**b**). Example of two axosomatic synapses where the orange axon contacts the blue soma. Two Gephyrin–Bassoon pairs (white arrows) are shown within the axonal bouton. **g–i** Single slice zoomed inset of cyan square in (**b**). Example of an axodendritic synapse where the red axon contacts the blue dendrite. White arrow points to the Gephyrin–Bassoon pair. **j–l** Single slice zoomed inset of yellow square shown in (**b**). Example of an axosomatic contact that lacks Gephyrin–Bassoon machinery (white arrow) and is not a true synapse. **m–o** Single slice zoomed inset of white square in (**b**). Example of an axon that is physically close, but does not actually form a synapse. White arrow points to the Gephyrin–Bassoon pair that corresponds to an axosomatic synapse between an unlabeled axon and the blue soma. **p–s** Normalized line intensity profiles of dotted lines drawn in **f**, **i**, **k**, and **o**, respectively. Distances between Gephyrin and Bassoon peaks or between two Brainbow cell membranes were measured and shown respectively for **p**, **q**, **s**, or **r**. Scale bars: (**c**) 10 μm (pre-expansion size). **d–o**) 1 μm (pre-expansion size). Expansion factor: (**b–o**) ~4×. See Supplementary Table 1 for more details.

color clusters (Supplementary Fig. 13). Within our dataset volume, the average upper and lower bound of unique PV neurons converging onto one PV neuron are estimated as $27.5 \pm 12.6$ and $10.9 \pm 4.5$ respectively ($n = 8$, mean ± standard deviation).

Next we shifted our attention from the distribution of axons to the distribution of synapses. We analyzed the distribution of PV–PV inhibitory synapses by splitting them into two spatial compartments: somatic and dendritic (Fig. 4f). We plotted the number of soma or dendrite targeting axons based on the number of putative synapses they provide. Cortical PV neurons are known as basket cells and tend to innervate the perisomatic region of other neurons[33]. While we confirmed the existence of putative PV–PV axosomatic synapses, a larger number was actually axodendritic. The soma targeting axons most commonly formed two putative synapses, but in contrast, the majority of the dendrite targeting axons formed one putative synapse. Figure 3d–f shows

an example of a single axonal bouton providing two putative axosomatic synapses. We then counted the total number of putative inhibitory synapses on the soma and plotted it alongside those that were annotated as PV–PV (Fig. 4g). We observed that ~$33.3 \pm 16.5\%$ of the somatic inhibitory synapses were PV–PV ($n = 8$, mean ± standard deviation). Because inhibitory synapses are plastic and can dynamically remodel[34–36], we wanted to determine if there were any differences in synapse size, a known correlate for synapse strength[37,38], between the 8 postsynaptic PV neurons. We looked at the distribution of Gephyrin volume for all PV–PV synapses for each of the 8 postsynaptic PV neurons and found they were largely consistent with no major differences (Fig. 4h). Next, we analyzed individual postsynaptic PV neurons and asked whether axons that provide more synapses are correlated with increased PSD sizes. Again, we observed PSD sizes were consistent and invariant of the number of synapses an axon formed (Supplementary Fig. 14).

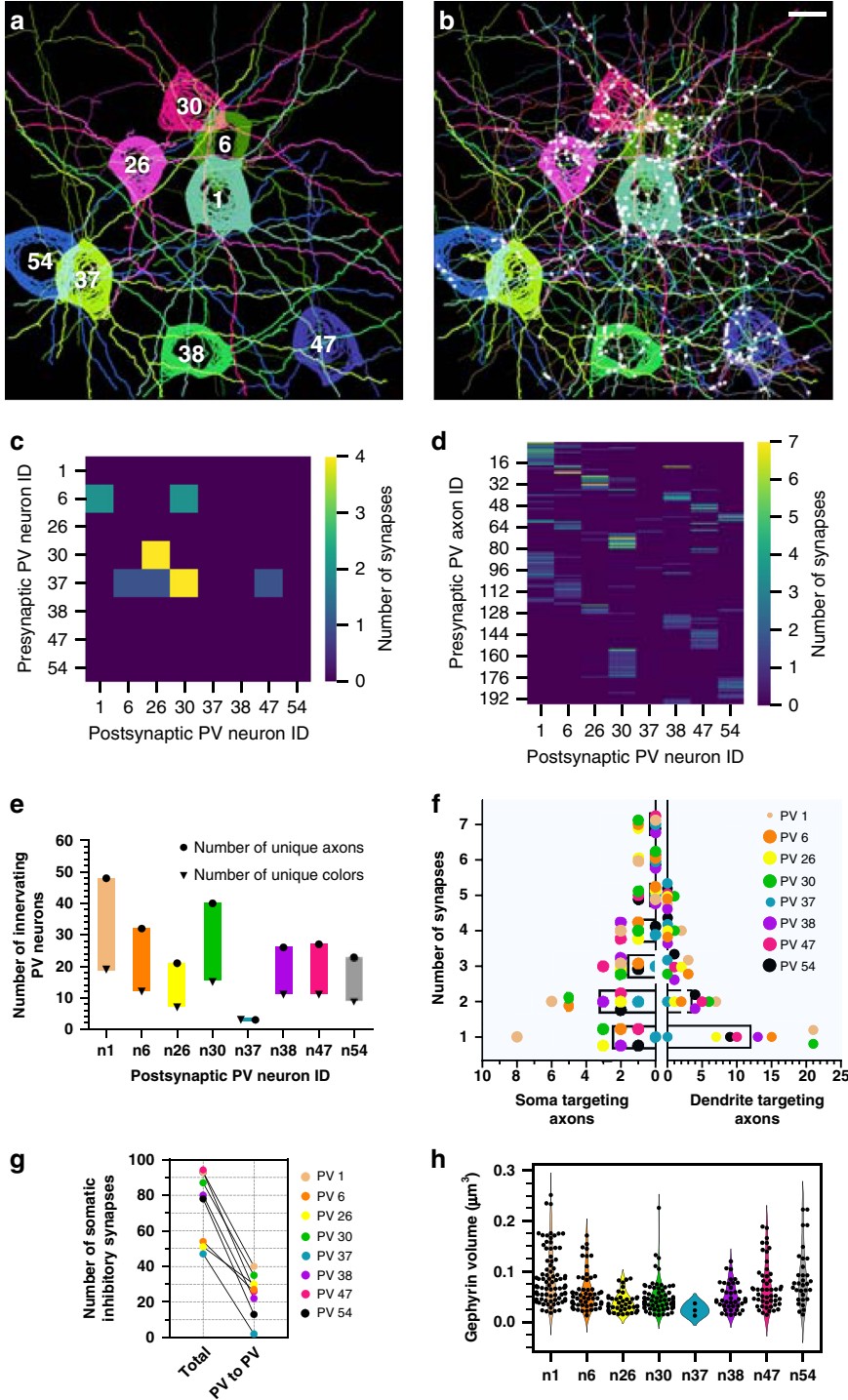

**Fig. 4 Molecularly specific analyses can be performed on spectral connectomics data. a** Eight Brainbow labeled PV neurons reconstructed with nTracer. Thick neurites represent dendrites; thin neurites represent axons. The neuron IDs are overlaid. **b** Same as **a**, plus all of the other 189 innervating axons overlaid and 422 identified synapses marked with white circles. **c** Connectivity matrix between the eight reconstructed PV neurons. **d** Connectivity matrix as in **c**, plus the 189 innervating axons. **e** Plot of the maximum and minimum number of unique presynaptic PV neurons that innervate each of the eight postsynaptic PV neurons. Circles represent the number of spatially distinct PV axons that formed synapses, while triangles represent the number of unique colors that could be identified from the innervating axons. **f** Distribution of the number of soma targeting and dendrite targeting axons as a function of the number of synapses they form. Each of the eight afferent PV neurons are color coded and plotted with the average represented by the bar graph. **g** Total number of somatic inhibitory synapses were plotted relative to the PV to PV subset for each of the eight postsynaptic PV neurons. **h** Inhibitory PSD (Gephyrin) size plotted for all the PV to PV synapses found on each of the eight postsynaptic PV neurons. Scale bars: 10 μm (pre-expansion size).

**Synaptic input maps as a metric to distinguish cell types.** Previous studies performing in situ synapse measurements tend to focus on excitatory neurons because dendritic spines are commonly used as a proxy for excitatory input[39]. Changes in spine size and density are often used as a correlate to structural synaptic plasticity[39]. Studies of inhibitory neuron synapses are more challenging and underrepresented since many of them are aspiny with no obvious morphological correlate. To meet this challenge, we simultaneously labeled aspiny BLA PV neurons with Brainbow, excitatory (Homer1), and inhibitory (Gephyrin) PSD markers (Fig. 5a). In round 1, three Brainbow FPs were immunostained and the sample was expanded ~4× and imaged (Fig. 5b). In round 2, Homer1 and Gephyrin were immunostained along with EYFP as a fiducial channel (Fig. 5c). ~4× Expansion of the sample gave us the resolution to optically resolve individual synaptic puncta along the Brainbow membrane (Fig. 5d–i). We reconstructed the dendritic morphology of 5 PV cells in a $220 \times 220 \times 85 \ \mu m^3$ imaging volume, and annotated all the excitatory ($915 \pm 274$, mean ± standard deviation, $n = 5$) and inhibitory ($409 \pm 141$, mean ± standard deviation, $n = 5$) PSDs to create a synaptic input map for each cell (Fig. 5j, k, Supplementary Movie 5). We propose that these synaptic input maps may be another useful metric to distinguish cell types. For example, we intriguingly observed that three out of five PV neurons possess a skewed distribution of excitatory vs. inhibitory inputs, while the other two PV neurons possess a more balanced distribution (Fig. 5l). The excitatory vs. inhibitory input ratio fundamentally influences a neuron's role in the circuit, and is another measure of connectivity that can be used to distinguish neuronal subtypes.

## Discussion

In summary, we present a LM based approach for connectivity analysis with molecular specificity, termed spectral connectomics, developed by combining Brainbow and miriEx technologies. We demonstrated multimodal measurements of morphology, molecular markers, and connectivity in a single mammalian brain section. We showed that miriEx supports robust preservation of antigens to enable multiple immunostaining, imaging, and stripping cycles. This allowed us to reconstruct the morphology and specify the molecular subtype of 53 different BLA interneurons in the same brain section. Adding endogenous immunostaining of pre and postsynaptic pairs enabled us to define putative synaptic connections between molecularly specified neurons. We traced 8 postsynaptic PV neurons, 189 innervating PV axons, and 422 PV–PV inhibitory synapses, and performed connectomic analyses similar to previous EM studies. Finally, we showed that Brainbow could be combined with endogenous PSD immunostaining to quantify inhibitory and excitatory synaptic input onto individual aspiny interneurons.

miriEx enhances the potential of using Brainbow for mapping neuroanatomy. Depending on the labeling density, diffraction-limited LM may be insufficient for tracing small caliber Brainbow neurites[8,26]. Expansion microscopy allowed us to resolve these neurites effectively for morphology and connectivity analysis. The membrane targeted FPs also proved to be an useful landmark when probing synaptic machinery. Both pre and postsynaptic membranes were required along with immunostaining for pre and postsynaptic machinery to identify putative synapses. In addition, membrane targeted FPs are shown to be better than cytosolic FPs, which may not diffuse as well, for depicting small subcellular structures[25]. Neuron arbors span long distances that cannot be completely captured in thin brain sections using standard histological techniques. We showed that miriEx, a hydrogel based clearing method, can be applied in 500 μm brain sections to enable homogenous immunostaining (Supplementary Fig. 15). Consequently, we anticipate that future optimization will allow compatibility with millimeter thick Brainbow brain sections to study more complete neuroanatomy.

Three challenges that remain with combining Brainbow and miriEx are the limited color diversity, dilution of membrane FP signal, and long experimental time spans that come from multiple rounds of immunostaining. Throughout our experiments, we immunostained for only three out of four possible FPs. We can already observe tens of easily distinguishable color barcodes, which can be exponentially increased in the future by probing more FPs. For example, five FPs were used in the recently developed Bitbow, a "digital" format of Brainbow[40]. Expansion of the sample is important for increasing imaging resolution, but comes at the cost of diluting antibody signal. For instance, ~4× expansion results in a ~64-fold reduction of antibody signal. As a result, axons can become dim and challenging to trace long distances. That said, we expect a more photon efficient volumetric imaging modality, such as light sheet microscopy, will yield better SnR under the same labeling conditions than confocal microscopy used in our study. Moving forward, it will be important to optimize FP expression along axonal membranes, as well as explore alternative signal amplification technologies, such as immuno-SABER[16]. Finally, we acknowledge that our multiplexed antigen detection strategy can be time consuming, especially if more immunostaining and imaging rounds are needed. For example, the experiment in Fig. 1b demonstrating 5 rounds of immunostaining, imaging, and stripping took 18 days. Future efforts can explore antibody-DNA barcoding strategies that offer faster multiplexed detection.

One important caveat to mention is that different neuronal cell types have various axon morphology properties that can be challenging to reconstruct. For example, glutamatergic cortical neurons can have smaller calibre axons, with tinier boutons, compared to the PV axonal arbors we reconstructed in this study. Although the concepts behind spectral connectomics can be applied broadly, Brainbow labeling (such as AAV titer and expression time) and imaging conditions (such as laser power and resolution) may need to be tailored for different subtypes of neurons.

A critical component of our spectral connectomics strategy is molecular specificity, which comes from two sources: endogenous protein immunostaining and choice of Cre-driver line. The ability to probe endogenous molecular markers with miriEx increases the flexibility of studying multiple neuronal cell types simultaneously. For example, future work can focus on using a broad transgenic line (e.g., *GAD2*-Cre) that can be molecularly specified by additional immunostaining (PV, SOM, VIP, etc). Furthermore, miriEx immunostaining can be extended to investigate the molecular content of synapses. Bassoon, Gephyrin, and Homer1 are only a subset of the synaptic machinery that can be probed, which includes neurotransmitter transporters, receptors, gap junctions, and ion channels. The rich library of Cre-driver lines can give us a genetic handle on studying specific cell types. Labeling a subset from the global population of neurons also helps to constrain and focus our connectivity analysis. For example, we only quantified and analyzed 422 PV–PV synapses out of ~60,000 inhibitory synapses in our dataset (~0.7% of the total). Because we asked molecularly specific, targeted questions of PV–PV connectivity, we reduced the burden of tracing neurons and annotating synapses to a scale that was manageable by manual curation (~40 human work hours).

Moreover, many Cre-driver lines are actually composed of different neuronal subtypes[41]. Our definitions of cell types are constantly changing with the introduction of new technologies. We have progressed from using unimodal criteria (e.g.,

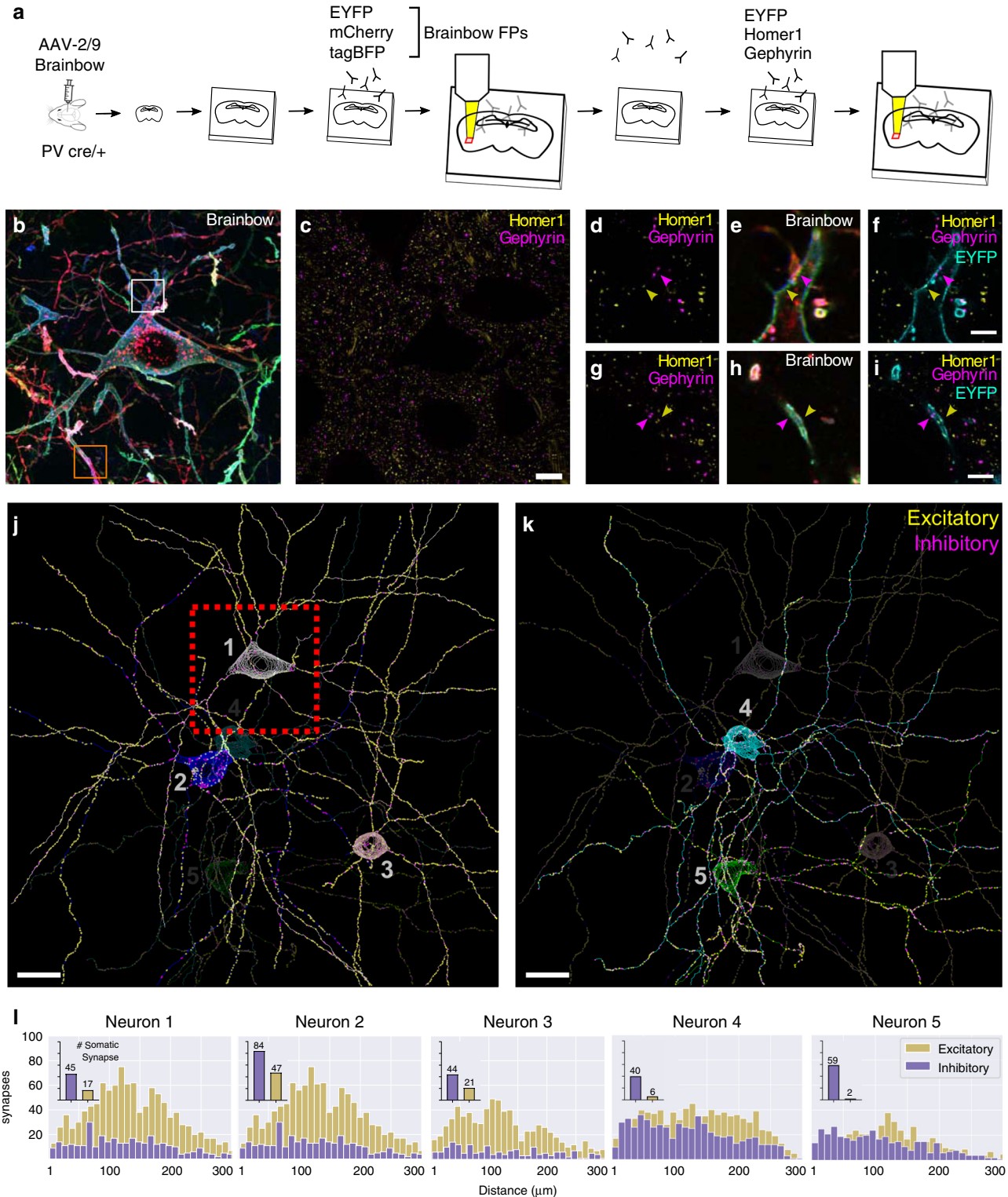

**Fig. 5 Putative excitatory and inhibitory synapses of aspiny inhibitory neurons can be quantified. a** Experiment design: Brainbow and endogenous PSD markers are imaged across two rounds of immunostaining using the EYFP channel for registration. **b** MIP of the three Brainbow channels imaged in round 1. **c** Single slice of excitatory (Homer1) and inhibitory (Gephyrin) PSDs imaged in round 2. **d–i** Zoomed in synaptic marker + Brainbow single slice images of the white square (top row) and orange square (bottom row) respectively. Yellow and purple arrows point to excitatory and inhibitory synaptic puncta respectively. **j**, **k** nTracer reconstruction of five neurons' morphology and putative synaptic inputs. The red box represents the field of view seen in **b**, **c**. **l** Histogram of the number of putative dendritic synapses as a function of dendritic length away from the soma for each neuron ($n = 5$ neurons from one experiment). The inset bar graph represents the number of putative somatic synapses for each neuron. Scale bars: (**c**) 10 μm (pre-expansion size), (**f**, **i**) 2.5 μm (pre-expansion size), (**j**, **k**) 20 μm (pre-expansion size). Expansion factor: (**b–i**) ~4×. See Supplementary Table 1 for more details.

morphology, physiology, or molecular expression) to using a combination of these properties in a multimodal fashion[42]. Recent studies have demonstrated the power of this multimodal approach to curate and refine definitions of inhibitory and excitatory neuronal cell types in visual cortex[43,44]. Similarly, our multimodal strategy for measuring morphology, molecular markers, and connectivity can be also applied towards refining cell-type definitions across the brain. More specifically, integrating connectivity information will enhance our understanding of how input/output properties inform cell-type definitions. In addition, spectral connectomics is compatible with techniques that measure physiological properties, such as patch clamp electrophysiology and in vivo calcium recordings, for more comprehensive analysis.

Finally, we envision spectral connectomics to supplement EM as an alternative strategy for mapping connectivity that is accessible to neurobiology labs. miriEx and Brainbow use commercially available, "off the shelf", reagents, and LM is a prevalent imaging modality across many labs. Data sizes are an order of magnitude smaller compared to EM datasets despite containing multiple channels (~20 vs. ~200 GB for a similar sized volume[5]). Targeted analysis makes it possible to map connectivity in a reasonable amount of time, and application of automated reconstruction and synapse segmentation methods will further simplify this process. EM based connectomics currently remains the gold standard for generating complete reference connectomes of animal brains, but we hope that spectral connectomics can be employed for validating molecularly specific circuit motifs and testing hypotheses of how neuroanatomy dynamically changes with perturbations. We imagine that in the future, spectral connectomics can be scaled up to generate comprehensive connectomes. This likely will come from developing new Brainbow designs to globally label neurons[30], whole brain expansion microscopy protocols, whole brain LM methods[45,46], whole brain immunolabeling techniques[47,48], and new computational pipelines for automated neuron reconstruction[49] and synapse segmentation[50]. Although challenging, many of these puzzle pieces are already being developed by the community. When combined in the spectral connectomics framework, they can fit together to reveal a mammalian connectome.

## Methods

**Mouse lines**. All experiments were carried out in compliance with the National Institutes of Health Guide for the Care and Use of Laboratory Animals. Our study protocol was reviewed and approved by the University of Michigan Institutional Animal Care & Use Committee. The transgenic mice used in this study were: PV-Cre (Jackson stock no. 008069), SOM-Cre (Jackson stock no. 013044), VGAT-Cre (Jackson stock no. 016962), Ai14 (Jackson stock no. 007914), and Thy1-YFP-H (Jackson stock no. 003782). P28–P56 male and female mice were used for virus injections. A total of ten mice on a C57/Bl6 genetic background were used in this study (two PV-Cre × SOM-Cre, three PV-Cre, one VGAT-Cre × Ai14, one PV-Cre × Ai14, one Thy1-YFP-H, two wild type).

**Brainbow AAV injections**. Brainbow3.0 AAV-2/9 and AAV-PhP.EB were obtained from Addgene and University of Michigan vector core, respectively. Transgenic mice (PV-Cre/SOM-Cre or PV-Cre) were anesthetized continuously with isofluorane and mounted on a stereotaxic frame. TagBFP-EYFP and mCherry-TFP virus were mixed together to reach a concentration of 1E12 gc mL$^{-1}$ individually, of which 500 nL was injected at 100 nL min$^{-1}$ using a capillary pipette backfilled with mineral oil at +3.5 ML and −1.7 AP relative to bregma, 2.7 DV from pia surface. Afterward, the pipette was left in place for 5 min. for the virus to diffuse, before slowly retracting out of the brain. We waited 3–4 weeks for virus expression before perfusing the animals. Brainbow3.0 AAV-PhP.EB was used for retro-orbital injection to systematically label neurons throughout the brain. Fifty microliters of mixed virus (1E12 gc total each for TagBFP-EYFP and mCherry-TFP) was injected into the retro-orbital sinus. We waited 3–4 weeks for virus expression before perfusing the animals.

**Mouse perfusion and tissue sectioning**. Mice were anesthetized with tribromoethanol and perfused transcardially with ice-cold 1× PBS, followed by 4% PFA in 1× PBS. The brains were dissected and postfixed in 4% PFA in 1× PBS

overnight shaking at 4 °C. The next day, brains were washed in 1× PBS before slicing on a vibratome (Leica VT1000s) at 100 or 200 μm thickness.

**miriEx protocol**. Acrylic acid N-hydroxysuccinimide ester (AAx, Sigma A8060) was prepared by dissolving in N,N-Dimethylformamide to 125 mM. Tissue samples were incubated with 1–5 mM Aax in a MBS buffer (100 mM MES, 150 mM NaCl, pH 6) with 0.1% Triton X-100 shaking overnight at 4 °C. The next day, the samples were washed 3× with 1× TBS (Bio-rad 1706435) for 1 h each to quench the reaction. The samples were then incubated in monomer solution (5.3% Sodium Acrylate, 4% Acrylamide, 0.1% Bis-Acrylamide, 0.5% VA-044, in 1× PBS) shaking overnight at 4 °C. The next day, the samples were gelled for 2.5 h at 37 °C in a humidity chamber by cover slipping the sample surrounded by monomer solution in a gelling chamber. The gel-tissue hybrids were then carefully cut out and denatured at 70 °C overnight in denaturing buffer (200 mM SDS dissolved in 1× TBS). The next day, denaturing buffer was washed out by incubating the samples 4× for 2 h each in 0.1% PBST (1× PBS with 0.1% Triton X-100) shaking at 50 °C.

**Immunohistochemistry of regular tissue sections**. For immunostaining of non-gelled samples, tissue sections were first blocked and permeabilized in Starting-Block (Thermo 37538) with 1% Triton X-100 shaking overnight at 4 °C. Tissues were washed the next day 3× with 1× PBS for 1 h each. Tissues were then incubated with primary antibodies diluted in 0.5% PBST for 3 days at 4 °C. After washing 3× with 0.5% PBST for 1 h each at RT, tissue sections were then incubated with secondary antibody diluted in 0.5% PBST for 2 days at 4 °C. Tissues were then mounted with Vectashield (Vectorlabs H-1000) and imaged. Primary and secondary antibody choice, concentrations, and incubation time can be found in Supplementary Table 1.

**Immunohistochemistry of gelled tissue sections**. Blocking and permeabilization of gel-tissue hybrids were skipped as they were found to minimally decrease background. Gel-tissue samples were incubated with primary antibody diluted in 0.1% PBST with 0.02% azide (PBSTz) at either RT or 37 °C. They were washed 3× with 0.1% PBST with 0.02% Azide at RT for 1 h each, and then incubated with secondary antibody diluted in 0.1% PBSTz at either RT or 37 °C. Primary and secondary antibody choice, concentrations, and incubation time can be found in Supplementary Table 1. Vendor information can be found in Supplementary Tables 2, 3.

**Gel expansion and fluorescence microscopy**. miriEx gel-tissue hybrids were incubated in 0.001× PBS three times for 45 min. each for saturated expansion to ~4×. They were then mounted in Poly-L-Lysine coated 6 cm dishes (Corning 354517) and submerged in 0.001× PBS. All confocal LM was performed using an upright Zeiss LSM780. Water immersion objectives (10×/NA0.4 or 20×/NA1.0) were directly lowered into the PBS solution over the sample for imaging. More imaging details are presented in Supplementary Table 1.

**Antibody elution**. miriEx gel-tissue hybrids that previously have undergone immunostaining, expansion, and imaging were shrunken by washing in 1× PBS 3× for 1 h each. They were then put in denaturing buffer (200 mM SDS dissolved in 1× TBS) at 70 °C overnight. Samples were washed the next day 4× for 2 h each in 0.1% PBST shaking at 50 °C before undergoing the next round of immunostaining.

**Human brain sample processing with miriEx**. Formalin-fixed human brain samples were obtained from the University of Michigan Brain Bank. A 1 cm slab of sensory cortex was macro dissected and washed in 1× PBS at 4 °C overnight. The sample was then sectioned into 100 μm slices before processing with miriEx.

**Image preprocessing**. Specific image processing steps for each imaging dataset are listed in Supplementary Table 1, and are briefly explained below.

Stitching of multi-tile datasets was performed using the BigStitcher[41] ImageJ/Fiji plugin. The dataset was loaded and converted to HDF5 format, and the tiles were arranged in the order they were imaged with 10% overlap. The Stitching Wizard was used to calculate pairwise shifts using phase correlation, verify links, and undergo global optimization. Affine refinement was then performed with a high threshold. The resulting interest points were used for non-rigid refinement during image fusion.

Chromatic aberration was corrected using the Detection of Molecules (DoM) ImageJ/Fiji plugin. To calibrate the DoM plugin, 0.5 μm TetraSpeck fluorescent beads were mounted on a slide and imaged on a Zeiss LSM780 confocal microscope. More specifically, we calibrated the orange channel (540–600 nm) excited with a 543 laser and far-red channel (630–700 nm) excited with a 633 laser to the green channel (480–540 nm) excited with a 488 laser. Calibration was performed for both the 10× and 20× objectives.

Histogram matching was done to normalize intensity between z-slices in image stacks using the nTracer Align-Master ImageJ/Fiji plugin. A high SnR z-slice (usually at the top or bottom of the stack) was chosen as the reference slice for which the rest of the stack was normalized to. All three Brainbow FP channels were histogram matched to the same reference slice.

**Image registration and alignment between rounds**. Following data preprocessing as described above, the fiducial marker channels from different rounds were loaded into ImageJ/Fiji Big Warp[51] plugin for rough, initial alignment. After this, the fiducial marker channels were registered through a B-spline transformation using Elastix[52,53]. The resulting transformation was applied to each individual channel to create a merged image hyperstack. To register and align images from different rounds that are different expansion sizes, the lower resolution fiduciary channel was upsampled using bilinear interpolation to match the voxel size of the higher resolution fiduciary channel. The two rounds were then registered and aligned as described above.

**Neuron reconstruction, synapse identification, and analysis**. nTracer, an ImageJ/Fiji plugin, were used to trace somas, dendrites, and axons of Brainbow labeled neurons (manual and tutorial videos can be found at https://www.cai-lab.org/ntracer-tutorial). The morphology reconstructions were exported in SWC format, and a custom Python script was used to render skeleton visualizations in TIFF format from SWC files. Putative synapses were identified and manually marked using the ROI manager in ImageJ/Fiji. The list of X, Y, Z synapse coordinates was saved to a CSV file and linked with their parent neurons by adding an additional data column in the SWC file marking synaptic locations along the dendrite or axon. Blender 2.81 (Blender Foundation; www.blender.org) or 3Dscript[54] was used to generate movies. Morphology features (i.e., number of stems, bifurcations, branches, etc.) were calculated by importing SWC files into Vaa3D's Global Neuron Feature plugin[55]. Sholl analysis was performed using ImageJ/Fiji Sholl Analysis plugin. Automatic segmentation of Gephyrin volumes was achieved through a custom Python script by first applying an Otsu thresholding step, followed by watershed segmentation of Gephyrin puncta. Plots were made in GraphPad Prism 8 or through Matplotlib Python scripts.

**PV immunostaining signal to noise quantification**. The average signal of ten randomly chosen PV somas was measured using Fiji/ImageJ across round 1, round 4, and round 7 PV immunostaining. The background noise in round 1, 4, and 7 was measured by averaging the signal of ten "empty holes" representing non-PV neurons. The SnR was calculated by dividing the signal of each neuron across the three rounds by the average background noise for that round.

**miriEx expansion measurement**. Hundred micrometers Thy1-YFP-H sections were mounted in Vectashield and imaged using confocal microscopy. The sections were then washed in 1× PBS to remove the vectashield and were processed with miriEx. An anti-GFP antibody was used to label native YFP signal. After miriEx, the samples were imaged in 1× PBS in the same location as before. They were then further expanded in 0.001× PBS (3× washes for 45 min. each) and imaged again. Elastix[2] was used to calculate an affine transformation between different expansion states, and the X, Y, and Z scaling factors were averaged to measure the expansion factor.

**Statistics and reproducibility**. The experiment in Fig. 1 was repeated independently one other time with similar results. The experiment in Fig. 2 was repeated independently one other time and shown in Supplementary Fig. 7. The experiment that generated imaging data for Figs. 3, 4, Supplementary Figs. 10, 11, 14 was repeated independently one other time with similar results and shown in Supplementary Fig. 14. Although not shown, the analysis for Figs. 13, 14 were repeated on the technical repeat shown in Supplementary Fig. 12 with similar efficiency. The experiment in Fig. 5 was repeated independently one other time with similar results. The experiment in Supplementary Fig. 2 involved three independent biological samples and was not repeated. The registration workflow shown in Supplementary Fig. 3 was repeated in all experiments involving multiple rounds of imaging. The experiments shown in Supplementary Figs. 3, 4, 8, 9, 15 were repeated independently one other time with similar results.

**Reporting summary**. Further information on research design is available in the Nature Research Reporting Summary linked to this article.

## Data availability
The data that support the findings of this study are available from the corresponding author upon reasonable request.

## Code availability
Custom code for analysis of image processing and data analysis is available from the corresponding author upon reasonable request.

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

## Acknowledgements

We thank Nigel Michki and Douglas Roossien for providing comments, editing, and discussion of the paper. We thank Harsimranjit Sekhon for insightful discussion. We thank Andy Lieberman for providing formalin-fixed human brain tissue. We thank Bo Duan and Lorraine Horwitz for providing *SOM*-Cre mice and genotyping assistance. We thank Scidraw.io and Luigi Petruco for vector graphics used in our figures. F.Y.S. acknowledges support by National Institutes of Health (NIH) 1F31NS11184701. D.C. acknowledges support by NIH 1UF1NS107659, 1R01MH110932, and 1RF1MH120005, and National Science Foundation NeuroNex-1707316. E.S.B. acknowledges support by Lisa Yang, John Doerr, the Open Philanthropy Project, and Schmidt Futures, and NIH 1R01NS102727, 1R01EB024261, and 1R01MH110932.

## Author contributions

F.Y.S. and D.C. conceived the project, designed the experiments, and wrote the paper with input from all authors. F.Y.S. performed and analyzed all miriEx and Brainbow experiments. H.P.J.C. and F.Y.S. traced the 53 BLA PV and SOM neurons. LAW wrote code for automatic segmentation of Gephyrin puncta and helped create the BLA molecular subtypes video. M.M.H. and F.Y.S. quantified Gephyrin–Bassoon and Homer1–Bassoon pairs, and PV and Brainbow overlap. E.S.B. supported ExM development. D.C. initiated and supervised the project.

## Competing interests

E.S.B. is a co-inventor on multiple patents related to expansion microscopy, and also co-founder of a startup related to expansion microscopy. The remaining authors declare no competing interests.
