## [Peer Review File · Nature Communications]

REVIEWER COMMENTS

Reviewer #1 (Remarks to the Author):

The authors propose an exciting and potentially very useful strategy to perform local connectivity analyses in brain slices. Using brainbow allows dense reconstructions of neurons and their morphologies, multiplexed molecular marker imaging allows cell classification and imaging of synaptic proteins identifies putative synapses. This novel approach can be carried out using regular confocal microscopy and therefore could be widely used by the neuroscience community. Researchers could study genetically identified cells, sub-classify them using molecular markers and identify local connectivity with cells in their close surrounding. It seems that antibodies working in Western blots have a high chance to also work with the miriEX approach, which is an attractive feature. The authors illustrate the power of their approach by analysing the connectivity of PV to PV cells and they identify the total number and dendritic distribution of excitatory and inhibitory synapses. The paper is well written and nicely illustrated.

Here are my main concerns:

1. Putative synapses

While the visualisations of synapses shown in this work are quite convincing, it seems that the improvement in resolution achieved by 4x expansion is not quite sufficient to unequivocally identify synapses. What does it help to identify candidate synapses? In order for the approach to become really useful, the authors need to demonstrate that the connections they show between two neurons are indeed synapses (or, at a minimum, determine the likelihood that a putative synapse is a real synapse that connects two identified cells). 3D analysis of the fluorescence data representing synaptic contacts might help to improve the confidence with which synapses can be reliably identified.

2. Accessible volume

The approach is introduced for sections of up to 200 μm thickness. Cortical principal neurons (or other large neurons such as cerebellar Purkinje neurons) can measure hundreds of micrometers with regard to the dendritic domain. Their axon forms recurrent projections that cover even larger volumes. Even when placed perfectly within the section, none of these cells can be analysed in their entirety using the approach described here. While interneurons have a much smaller dendritic domain, their axon typically occupies a much larger volume. Even for cortical interneurons it might be difficult to capture entire cells (Fig. 4 seems to show only incomplete cells). Hence, the approach will only work on rather small neurons and is therefore limited (for the full connectomics analysis to $100 \times 100 \times 60 \mu\text{m}^3$). Could also slices much thicker than 200 μm be used to alleviate this limit? Is fluorophore bleaching induced by confocal imaging of large volumes a limiting factor?

3. Directly versus indirectly connected neurons?

An issue related to point 2 is that most of the physiologically meaningful synaptic connections within ensembles/assemblies of neurons are indirect, meaning that neuron 1 contacts neuron 2 and changes its firing properties, and neuron 2 in turn contacts neuron 3 and impacts its function. Could such connections be resolved with the presented approach?

4. Tracing precision and reliability

How precise is the tracing of the labeled neurons? How well does the spectral separation work? Unfortunately, the cited paper Roossien et al. did not provide much information on that. Often fluorescently labeled structures are interrupted by short non-labelled stretches. How was this taken into consideration? Can the approach reliably differentiate dendrites from axons? Can axons measuring $<100 \text{ nm}$ be detected reliably? While the reconstructed cells look quite convincing, it remains unclear to which extent the traced neurons correspond to the 'real' neuron as reflected by the fluorescence signal. The traced cells should be more transparently validated.

5. Number of repetitions

How many times were the experiments shown in Figs. 2-4 repeated? Are these n=1 experiments? Would a repetition of the same experiment yield the same results? Reproducibility is of key importance for new methods and needs to be explicitly demonstrated.

Further comments and concerns:

26: what are "network morphologies"?

109-119: The authors performed the very important control of showing repetitive re-staining with the same antibody. However, a quantitative analysis should be added (shown as a graph intensity versus staining round). Fig.1b seems to show quite a strong reduction in the signal, suggesting that a substantial loss of antigens occurs. This limits the conclusion that "antigen signal can be reliably retrieved" and needs to be pointed out in the text.

118: 3-4 antigens - Fig.S5 shows 3 antigens

137: ... were processed with miriEx => mention that 4x expansion factor was used. In general, it is difficult to trace which of the work was done at what expansion factor. The expansion factor should be mentioned in each legend. Table S1 should be referred to from the legends of all figures and supplementary figures. At present, Table S1 is only referred to in the Methods. I only discovered the table after I read most of the manuscript, in fact, it is very useful (however, a file name like "248001_0_supp_846324_q687bp" does not really draw much attention - why not name it Supplemental Table 1?). The table gives information about the expansion factor indirectly by listing the concentration of PBS. What is the expansion factor corresponding to 0.01 PBS?

146: Fig. 2g-h => f-h

147: Fig. 2l-m => k-m

179-181: the sample was expanded ... was again expanded - does that mean: first 2x expansion, then 4x expansion - how are the differently expanded datasets registered?

Fig. 2:

- Mention the number of confocal image planes (and the volume (z distance) they covered) that were used to generate the MIP. Concerning the whole manuscript, this information should be added to column "imaging" of Table S1 for all figures.
- Cells shown within the labelled square in panel f does not seem to match with the cells shown in b-e. Show the correct ROI matching with the confocal panels

Fig. 3:

- Colour scheme in panel k?

Fig. S1:

- "The sample was then stripped using heat/SDS and reimaged". Was the same procedure used as described in methods under "Antibody elution"?
- n=3 samples => specify what samples means (three different sections from the same animal?)
- Does the elution work equally well as shown for anti-mCherry 488 when using other antibodies, for example such that have a higher binding affinity?

Fig. S2: 4x expansion is quite limited in terms of the gain in resolution that can be attained. A higher gain would resolve synaptic structures better, so that synaptic connections could be reliably detected. Did the authors explore higher expansion factors and is 4x really the best compromise between the gain in resolution and detectability of the signal in the expanded state? At least Fig. 5 used >4x expansion (PBS 0.0001x). Why was the connectivity analysis not done with 0.0001x?

Fig. S4:

- DAPI channel: clip the intensity range of the image such that the nuclei are better visible
- What was the rationale of using 5 mM Aax in this example, rather than 1 mM as in most other experiments?

Fig. S7:

- Scale bar 5 nm? If correct, the synapses would be < 1nm, which is not possible.
- Are the number of synapses plausible in comparison to known synapse densities, considering that not all neurons are labeled?

Methods

- Animals - how many mice were used in the entire study?
- 28: brians => brains
- 53-59: Add the incubation time of primary and secondary antibodies.

How much time does a multiplexing experiment involving 5 staining rounds require? Mention in the text that the procedure is actually very time-consuming. To what extent can the analysis be automated?

Reviewer #2 (Remarks to the Author):

Connectomics analysis is a crucial step toward mapping whole neural circuitry with a synaptic level. EM is a current gold standard, while issues in throughput, difficulty in molecular interrogation, and scalability should be overcome. Here, the manuscript by Shen et al. reported a light microscopy approach with multiplex immunostaining and Brainbow toward connectivity analysis of defined cell types (spectral connectomics), named miriEx (a multi-round immunostaining Expansion Microscopy). The reviewer was convinced of the importance, the design of experiments, and most of the conclusions. Although the manuscript is almost at a publishable level, the reviewer recommends a few revisions for further improvement.

Major concerns:

- #1) Figure 3f, j/page 5, line 193-195: please clarify the reason of why the Bassoon-Gephyrin pair does not form a synapse. Does the slightly larger gap (135 nm in j vs. 110-120 nm in h and i) provide a sufficient reason? Another concern is that a slight distortion during ExM could bias the gap distance. Is there any evidence to rule out that possibility?
- #2) Mouse lines: In which experiment did the authors use Thy1-YFP-H strain? As long as the reviewer thoroughly read the manuscript, it might be used in the experiment in Supplementary Fig. 2. But there is no description regarding the mouse strain and the source of signals in neurons.
- #3) On page 3, line 96-97: SWITCH method (Murray et al. Cell 163, 1500–1514, 2015) intensively tested multiplex immunostaining with adopting a similar stripping-restaining strategy. The reviewer recommends to add it among the references.

Minor concerns

In Fig. 1b, information on which round of PV image was used in Merge panel should be clarified.

On page 3, line 121-127: please indicate the corresponding figure panel(s).

On page 3, line 146: the figure reference (Fig. 2g-h) is apparently corresponding to Fig. 2j, k.

Please provide more detailed descriptions in the legends of Supplementary Fig. 3 and 4.

In Fig. 4e, Is the bar for n37 deliberately turned sideways?

Please thoroughly check typos in the manuscript, such as: "... doesn't actually actually form a synapse..." (Fig. 3f legend); "brians were washed..." (Mouse perfusion and tissue sectioning in the Method section); AAX/Aas (miriEx protocol in the Methods section).

Reviewer #3 (Remarks to the Author):

This clearly written and beautifully illustrated manuscript describes several important individual innovations and combinations of same likely to be widely useful in many areas of neuroscience, including, but not limited to the specific focus area, "connectomics", called out in the present manuscript. I have no qualms about recommending publication in essentially the present form. I would suggest, however, that the writing should be clarified in two ways. (1) The term "dense" in the context of connectomic reconstruction is now coming to be understood in a fairly absolute sense to refer to reconstructing EVERY neuron. In this writing, it is used in a much more relative sense (i.e., to labeling and reconstructing at most all GABAergic neurons - a minor fraction of the total neuron population). The text should be modified to avoid this clash of meanings. (2) A related caveat might also be in order: GABAergic neurons tend to have shorter and thicker axons and larger boutons, making them easier to identify spectrally and trace than the thinner and therefore optically "noisier" axons and boutons of the more populous glutamergic neurons. There should thus be a disclaimer that what is shown to be true here for GABA neurons might not be true for those generally much longer and skinnier glutamatergic axons and their often-tinier boutons. Having said that, the present report demonstrates methods of great promise, even if it did turn out to be as useful as claimed here only for GABAergic neurons. In any case, some discussion of this issue would certainly make this important contribution even more valuable.

Stephen J Smith

REVIEWER COMMENTS

Reviewer #1 (Remarks to the Author):

The authors propose an exciting and potentially very useful strategy to perform local connectivity analyses in brain slices. Using brainbow allows dense reconstructions of neurons and their morphologies, multiplexed molecular marker imaging allows cell classification and imaging of synaptic proteins identifies putative synapses. This novel approach can be carried out using regular confocal microscopy and therefore could be widely used by the neuroscience community. Researchers could study genetically identified cells, sub-classify them using molecular markers and identify local connectivity with cells in their close surrounding. It seems that antibodies working in Western blots have a high chance to also work with the miriEX approach, which is an attractive feature. The authors illustrate the power of their approach by analysing the connectivity of PV to PV cells and they identify the total number and dendritic distribution of excitatory and inhibitory synapses. The paper is well written and nicely illustrated.

We thank reviewer #1 for the kind comments and have addressed the concerns here and revised the manuscript in the result and discussion sections (highlighted in blue).

Here are my main concerns:

1. Putative synapses

While the visualisations of synapses shown in this work are quite convincing, it seems that the improvement in resolution achieved by 4x expansion is not quite sufficient to unequivocally identify synapses. What does it help to identify candidate synapses? In order for the approach to become really useful, the authors need to demonstrate that the connections they show between two neurons are indeed synapses (or, at a minimum, determine the likelihood that a putative synapse is a real synapse that connects two identified cells). 3D analysis of the fluorescence data representing synaptic contacts might help to improve the confidence with which synapses can be reliably identified.

We thank the reviewer pointing out this important limitation for all microscopy methods that attempt to identify putative synapses. Historically, ultrastructural features from electron microscopy (EM) images (i.e. synaptic vesicles, postsynaptic density, and synaptic cleft) have been used to define synapses. Recent advances in super resolution light microscopy have demonstrated that pre- and postsynaptic proteins themselves can provide an alternative definition of a synapse (Sigal et al. 2015, Cell, Gao et al. 2019, Science) given we have enough resolution to resolve the synaptic pair. The distance between Gephyrin Bassoon pairs has been reported to be 100-200 nm (Chang et al. 2017, Nature Methods). Assuming the average distance is ~150 nm, 4x expansion enables confocal imaging resolutions of $75 \times 75 \times 200 \text{ nm}^3$ which sufficiently meets Nyquist sampling to resolve Gephyrin Bassoon pairs in the lateral direction. Our work pushes this further to include the Brainbow-colored membranes between the pre- and postsynaptic sites as part of the trio-structure, similar to EM does, as a stringent criteria

when making determination of putative synapses. We do recognize that our axial resolution is insufficient to resolve Gephyrin Basson pairs that are aligned vertically in space. We appreciate the reviewer's suggestion of 3D analysis and added an additional supplemental movie (movie S3) that shows a 3D view of a synapse.

We agree that it would be ideal to verify the synapses identified with spectral connectomics directly with correlated EM imaging. However, it is beyond the scope of this manuscript as EM is not compatible with expansion protocol due to membrane extraction. That said, if multiple putative synapses are identified between the exact two neurons, from a statistics perspective, it is more likely that these two are functionally connected. We believe it is very helpful to identify potential connections between two neurons unequivocally. For instance, one can Brainbow-label two types of genetically determined neurons in, such as PV-Cre and SST-Cre driver mice. Spectral connectomics analysis can then help rapidly screen putative connections across multiple brain regions of interest. Once the putative synapses are identified in particular brain regions, follow-up validation experiments can be performed.

2. Accessible volume

The approach is introduced for sections of up to 200 μm thickness. Cortical principal neurons (or other large neurons such as cerebellar Purkinje neurons) can measure hundreds of micrometers with regard to the dendritic domain. Their axon forms recurrent projections that cover even larger volumes. Even when placed perfectly within the section, none of these cells can be analysed in their entirety using the approach described here. While interneurons have a much smaller dendritic domain, their axon typically occupies a much larger volume. Even for cortical interneurons it might be difficult to capture entire cells (Fig. 4 seems to show only incomplete cells). Hence, the approach will only work on rather small neurons and is therefore limited (for the full connectomics analysis to $100 \times 100 \times 60 \mu\text{m}^3$). Could also slices much thicker than 200 μm be used to alleviate this limit? Is fluorophore bleaching induced by confocal imaging of large volumes a limiting factor?

The reviewer has pointed out one of the problems with mapping neuron morphologies: neurons span large dimensions that are difficult to capture completely. Tissue clearing techniques, like CLARITY and CUBIC, render thick pieces of intact tissue transparent to capture the whole morphology of neurons. We would like to make three points. First, the vast majority of neuron morphology studies have been performed in 300 μm thick sections or thinner, relying on biotin filling after patch clamp recording. Although our understanding of neuron morphologies has been incomplete, valuable biological insights have still been gained by sampling only the proximal portion of neurons. Second, recent mouse EM studies (Motta et al. 2019, Science, Schneider-Mizell et al. 2020, bioRxiv) have mapped roughly the same volumetric space as we do and have generated new biological insights, hinting that spectral connectomics of similar sized volumes can still be useful. Third, we have successfully applied miriEx to thicker brain sections up to 500 μm , which permits capturing more morphology and connections in intact tissue. We purposefully chose a post-gelation immunostaining expansion microscopy strategy to facilitate antibody labeling of thick brain sections. We added an additional supplemental figure 14 to demonstrate homogenous tdTomato antibody staining across a 500 μm thick brain section

using miriEx. We also added the following text to the discussion on page 8: “Neuron arbors span long distances that cannot be completely captured in thin brain sections using standard histological techniques. We showed that miriEx, a hydrogel based clearing method, can be applied in 500 μm brain sections to enable homogenous immunostaining (**Fig. S15**). Consequently, we anticipate that future optimization will allow compatibility with millimeter thick Brainbow brain sections to study more complete neuroanatomy”

Thicker miriEx brain sections come with imaging challenges. As the reviewer pointed out, thicker samples necessitate longer imaging times by confocal imaging and would induce fluorophore bleaching. This is partly mitigated by using fluorescent dyes that are more photostable than native emission from fluorescent proteins themselves. In the future, one can also employ lightsheet imaging of thicker miriEx brain sections to reduce photobleaching as well as increase imaging speed.

3. Directly versus indirectly connected neurons?

An issue related to point 2 is that most of the physiologically meaningful synaptic connections within ensembles/assemblies of neurons are indirect, meaning that neuron 1 contacts neuron 2 and changes its firing properties, and neuron 2 in turn contacts neuron 3 and impacts its function. Could such connections be resolved with the presented approach?

Yes, the reviewer points out a valuable goal of connectomics of mapping specific circuit motifs. We anticipate that indirect synaptic connections can be identified using spectral connectomics. For example, in our data we saw that PV neuron 37 connects with neuron 1 indirectly through intermediate neuron 6. To highlight the importance of the reviewer's suggestion, we added the following line to the text on page 5-6:

“We did observe an example of indirect connectivity where neuron 37 connects with neuron 1 indirectly through neuron 6, highlighting that neuron assemblies can be mapped with spectral connectomics.”

4. Tracing precision and reliability

How precise is the tracing of the labeled neurons?

As pointed out in the cited paper Roossien et al., the inter-user error (one important perspective of tracing precision) depends on image resolution and less on (color) complexity. This is similar to the reconstruction precision in EM images that if sufficient resolution is achieved, even saturated labeling can be reliably reconstructed. As a human-guided tracing strategy, nTracer generates traces to sequentially connect user defined anchor points along the neurites. The tracing precision of a particular result mostly depends on how well the tracer distinguishes the neurons in colors to correctly position the anchor points. nTracer can then fill in the “eye-traced” smooth paths between two anchor points automatically by a few mouse/keyboard hits to greatly shorten the tracing time. nTracer uses A*, a classical minimum cost pathfinding algorithm, that generates very reliable paths given that the two consecutive anchor points are close enough

with each other. We added Supplemental Figure 11 to show tracing overlaid on top of raw image for the reviewers and readers to determine the tracing precision of Figure 4.

How well does the spectral separation work? Unfortunately, the cited paper Roossien et al. did not provide much information on that.

The reviewer raised a very important point that the spectral separation is critical for tracing accuracy, not only for human to distinguish neurons but also for A* to generate reliable paths. First of all, we utilized spectrally very separable dyes in the miriEx protocol to make sure there is no crosstalk between Brainbow spectral channels. Secondly, to achieve the best tracing results, nTracer integrates a channel alignment function and a histogram normalization function (Formula 1 of Roossien et al. 2019, Bioinformatics) to process the whole raw image stack. This post-acquisition step corrects optical aberrations and photobleaching to not only generate consistent spectral distribution and intensity across the whole imaging depth, but also allows best color separation (Supplemental Figure 2. of Roossien et al. 2019, Bioinformatics). Therefore, users can normally quite confidently follow a neuron throughout the whole image stack guided by relatively consistent color. Formula 2 of the cited paper Roossien et al. gives the color definition used for A* cost computation, which combines an α -weighted spectral term and a β -weighted intensity term. This gives the flexibility and tracing consistency when searching along neurites in a region of high dynamic range. Taking together, feedbacks from more than 30 users of nTracer indicate that it generates very reliable tracing paths, as long as the user can “eye-trace” the data. Several real-time tracing demonstration videos have been posted on <https://www.cai-lab.org/ntracer-tutorial>, specifically https://www.youtube.com/watch?time_continue=157&v=2YqyxzAriQM&feature=emb_logo and https://www.youtube.com/watch?time_continue=21&v=KxieNK2HEP0&feature=emb_title.

Often fluorescently labeled structures are interrupted by short non-labelled stretches. How was this taken into consideration?

We agree with the reviewer that fluorescent labeling of neurons, especially using cytoplasmic fluorophores, are interrupted by short non-labeled stretches. This is caused by the large volume difference between the large and small calibre neuronal processes, which in turn caused high dynamic range of fluorescent signal between the two structures that an imaging system may not record. Membrane-tag, such as the farnesylation motif used in Brainbow AAVs, has been shown to label the small calibre structures including the spine neck and thin axon (Cai et al. 2013, Nature Methods). In addition, the seemingly broken short stretches in our images actually have low intensity fluorescence that is not obvious to human eyes but is above background to computer vision. nTracer also uses a window to display the local maximum intensity projection image, which helps the user to determine whether to connect the short “non-labeled” stretches based on the overall structure of the colored neurite. Once the decision is made, A* still reliably connects the “broken” parts using the spectral information in the low intensity stretches.

Can the approach reliably differentiate dendrites from axons?

Yes. There are two features that help us to reliably differentiate dendrites from axons: 1) dendrites normally have larger calibres which are shown as halo structures in the membrane cross sections; 2) our imaging resolution is high enough that the Gypherin postsynaptic staining

and the Bassoon presynaptic staining appears “inside” the membrane of dendrites and axons, respectively, and can be used to distinguish them.

Can axons measuring <100 nm be detected reliably?

In principle, the answer is “yes” as long as the labeling is bright enough. That said, very small axons, such as those of the striatal cholinergic neurons, appear as threads 1-2 pixels wide. If labeling is weak, it becomes difficult to trace in dense regions.

While the reconstructed cells look quite convincing, it remains unclear to which extent the traced neurons correspond to the ‘real’ neuron as reflected by the fluorescence signal. The traced cells should be more transparently validated.

We now added a Supplemental Figure 11 to show tracing overlaid on top of raw image for the reviewers and readers to determine the tracing precision of Figure 4. Several real-time tracing demonstration videos have been posted on <https://www.cai-lab.org/ntracer-tutorial> (specifically https://www.youtube.com/watch?time_continue=157&v=2YgyxzAriQM&feature=emb_logo and https://www.youtube.com/watch?time_continue=21&v=KxieNK2HEP0&feature=emb_title), which shows the reliability of A*-based tracing.

5. Number of repetitions

How many times were the experiments shown in Figs. 2-4 repeated? Are these $n=1$ experiments? Would a repetition of the same experiment yield the same results? Reproducibility is of key importance for new methods and needs to be explicitly demonstrated.

The sample prep experiments in Figures 2-4 were each repeated multiple times. To demonstrate reproducibility of the technique, we added Supplemental Figure 7 as a technical replicate of Figure 2, and Supplemental Figure 12 as a technical replicate of Figures 3-4. We did not go as in depth with the analysis as we did with Figs. 2-4 because we are not trying to make definitive biological claims. The goal of Figs. 2-4 is to show an example of spectral connectomics and highlight potential analyses one can perform.

Further comments and concerns:

26: what are “network morphologies”?

We apologize for the vague term and replaced “network morphologies” with the “morphologies” in the manuscript.

109-119: The authors performed the very important control of showing repetitive re-staining with the same antibody. However, a quantitative analysis should be added (shown as a graph intensity versus staining round). Fig. 1b seems to show quite a strong reduction in the signal, suggesting that a substantial loss of antigens occurs. This limits the conclusion that “antigen signal can be reliably retrieved” and needs to be pointed out in the text.

We thank the reviewer for making this point. We quantified the signal to noise of the same 10 randomly selected somas across r1, r4, and r7 PV staining and added the plot as **Fig 1c**.

We removed the sentence on page 3: “We found that the same parvalbumin (PV) antibody showed similar staining patterns across multiple rounds with high signal remaining in neurites, indicating that antigen signal can be reliably retrieved, even in later rounds of miriEx” and replaced it with “We quantified the signal to noise (SnR) of PV immunostaining in rounds 1, 4, and 7 and found that there was a larger decrease from round 1 to round 4 than from round 4 to round 7 (Fig 1c). This suggests that the amount of protein lost in the antibody elution step decreases after every round. The fact that round 7 PV SnR is still one third of round 1 PV SnR even after 6 rounds of elution indicates that antigens are still preserved in later rounds of miriEx and can be retrieved.”

118: 3-4 antigens - Fig.S5 shows 3 antigens

We apologize for the confusion. For Fig. S5, 3 antibodies were used in the first 4 rounds of immunostaining and 2 antibodies were used in round 5. DAPI was co-stained in each round to use for registration and alignment. We clarified the text to now read 2-3 antigens.

137: ... were processed with miriEx => mention that 4x expansion factor was used. In general, it is difficult to trace which of the work was done at what expansion factor. The expansion factor should be mentioned in each legend. Table S1 should be referred to from the legends of all figures and supplementary figures. At present, Table S1 is only referred to in the Methods. I only discovered the table after I read most of the manuscript, in fact, it is very useful (however, a file name like “248001_0_supp_846324_q687bp” does not really draw much attention - why not name it Supplemental Table 1?). The table gives information about the expansion factor indirectly by listing the concentration of PBS. What is the expansion factor corresponding to 0.01 PBS?

We thank the reviewer for this point. Estimated expansion factors are now mentioned in each legend and Table S1 is referred to as well. We apologize for the typo; 0.01x PBS in Table S1 was corrected to 0.001x PBS. We added the estimated expansion factors into Table S1.

146: Fig. 2g-h => f-h

We thank the reviewer for pointing out this error. We corrected the figure reference to 2j-k to point readers to the morphology reconstructions.

147: Fig. 2l-m => k-m

We are confused by what the reviewer meant. Fig. 2l-m is the correct figure reference to point the readers to the morphological quantifications.

179-181: the sample was expanded ... was again expanded - does that mean: first 2x expansion, then 4x expansion - how are the differently expanded datasets registered?

We apologize for the confusion here. In the specific sentence the reviewer is referring to, we meant that the sample was expanded to ~4x the first round, shrunken, and then expanded again to ~4x the second round. We clarified this in the text.

However, later on for supplemental figure 9, we performed a third round of molecular marker immunostaining for somatostatin. In this case, the sample was imaged at ~2x expansion since higher resolution is not required for resolving molecular markers. The expansion states are reflected in the **Fig. S9a** in the size of the gel schematic. One can align and register images from different rounds that are different expansion sizes using the same pipeline described in Supplemental Figure 3 and methods. Either the lower expansion image can be upsampled to match the voxel size of the higher expansion image, or vice versa. In this case, we chose the former strategy.

We added the following description to the methods section on page 4: “To register and align images from different rounds that are different expansion sizes, the lower resolution fiduciary channel was upsampled using bilinear interpolation to match the voxel size of the higher resolution fiduciary channel. The two rounds were then registered and aligned as described above.”

Fig. 2:

- Mention the number of confocal image planes (and the volume (z distance) they covered) that were used to generate the MIP. Concerning the whole manuscript, this information should be added to column “imaging” of Table S1 for all figures.

We thank the reviewer for this point. The number of confocal image planes and Z distances for all the MIPs from Fig 2, 3, 5, and sFig9 were added to column “imaging” of Table S1 for all figures.

- Cells shown within the labelled square in panel f does not seem to match with the cells shown in b-e. Show the correct ROI matching with the confocal panels

We apologize for the confusion. Cells in panel 2f do match the cells shown in b-e. To clarify this point, we adjusted the brightness/contrast of panels f-i and added the corresponding numerical cell labels for cells 2, 4, 5.

Fig. 3:

- Colour scheme in panel k?

We adjusted the color scheme in panel 3k to be more consistent with the rest of the figure.

Fig. S1:

- “The sample was then stripped using heat/SDS and reimaged”. Was the same procedure used as described in methods under “Antibody elution”?

Yes, the procedure is the same as described in methods under Antibody elution. To avoid confusion, we added this description into the figure legend for Figure S1.

- n=3 samples => specify what samples means (three different sections from the same animal?)

We added additional description into the figure legend to clarify this point: "n=3 separate samples from 1 animal".

- Does the elution work equally well as shown for anti-mCherry 488 when using other antibodies, for example such that have a higher binding affinity?

Yes, the elution seems to be very effective across a wide range of antibodies that we have tested. We have never encountered issues of "leftover signal" from a previous round of immunostaining when using the elution method that we describe in the methods.

Fig. S2: 4x expansion is quite limited in terms of the gain in resolution that can be attained. A higher gain would resolve synaptic structures better, so that synaptic connections could be reliably detected. Did the authors explore higher expansion factors and is 4x really the best compromise between the gain in resolution and detectability of the signal in the expanded state? At least Fig. 5 used >4x expansion (PBS 0.0001x). Why was the connectivity analysis not done with 0.0001x?

We thank the reviewer for pointing out this error. 0.0001x PBS was corrected to 0.001x PBS in Table S1. As the reviewer points out, a higher expansion factor would be beneficial for resolving synapses, but has the consequence of diluting antibodies signal and making them harder to detect. Based on our experience, we believe ~4x expansion offers a nice balance of maintaining enough antibody signal and being able to resolve synapses. We did not try higher expansion factors using our current confocal imaging setup because we worried that lower signals would require higher laser powers and worsen photobleaching issues.

Fig. S4:

- DAPI channel: clip the intensity range of the image such that the nuclei are better visible

- What was the rationale of using 5 mM Aax in this example, rather than 1 mM as in most other experiments?

We thank the reviewer for this point and adjusted the brightness/contrast settings to make the nuclei more visible.

The rationale for using 5 mM was that the DAPI channels shown were taken from the 7 round miriEx imaging dataset shown in **Fig 1b** that happened to use 5 mM.

Fig. S7 (now changed to Fig. S8):

- Scale bar 5 nm? If correct, the synapses would be < 1nm, which is not possible.

We thank the reviewer for pointing out this error. We corrected the scale bar to read 5 μ m.

- Are the number of synapses plausible in comparison to know synapse densities, considering that not all neurons are labeled?

We think our estimates of synapse densities are plausible because the antibodies used were specific to endogenous synaptic proteins. A recent EM study (Motta et al. 2019, Science), which found ~400,000 total synapses within a ~500,000 μm^3 volume, resulting in estimated synapse density of 0.8 synapses/ μm^3 . This is close to our measurement of 1.28 synapses/ μm^3 , and the difference can be reasonably attributed to sampling differences like brain region, animal age and sex.

Methods

- Animals - how many mice were used in the entire study?

9 mice were used in the entire study. The following text was added to “Mouse lines” section under methods: A total of 10 mice on a C57/Bl6 genetic background were used in this study (2 PV-Cre x Som-Cre, 3 PV-Cre, 1 VGAT-Cre x Ai14, 1 PV-Cre x Ai14, 1 Thy1-YFP-H, 2 wildtype).

- 28: brians => brains

We have corrected this spelling mistake.

- 53-59: Add the incubation time of primary and secondary antibodies.

The incubation time of primary and secondary antibodies can be found in Supplemental Table 1.

How much time does a multiplexing experiment involving 5 staining rounds require? Mention in the text that the procedure is actually very time-consuming. To what extend can the analysis be automated?

The reviewer makes a good point that multiple rounds of miriEx can be time consuming. 5 rounds would take a total of ~18 days (3 days to crosslink, gel, and denature the sample, and then 3 days for every round of immunostaining, imaging, and stripping). We added the following text to the discussion:

“Finally, we acknowledge that our multiplexed antigen detection strategy can be time consuming, especially if more immunostaining and imaging rounds are needed. For example, the experiment in **Fig 1b** demonstrating 5 rounds of immunostaining, imaging, and stripping took 18 days. Future efforts can explore antibody-DNA barcoding strategies that offer faster multiplexed detection.”

The analysis has high potential to be automated in the future, especially the neuron reconstruction and synapse annotation. Computational methods have already been developed for EM that automatically segment neurons using flood filling networks (Januszewski et al. 2018, Nature Methods) and synapses (Janelia Fly EM). We imagine that these same methods can be adapted for spectral connectomics light microscopy datasets.

Reviewer #2 (Remarks to the Author):

Connectomics analysis is a crucial step toward mapping whole neural circuitry with a synaptic level. EM is a current gold standard, while issues in throughput, difficulty in molecular interrogation, and scalability should be overcome. Here, the manuscript by Shen et al. reported a light microscopy approach with multiplex immunostaining and Brainbow toward connectivity analysis of defined cell types (spectral connectomics), named miriEx (a multi-round immunostaining Expansion Microscopy). The reviewer was convinced of the importance, the design of experiments, and most of the conclusions. Although the manuscript is almost at a publishable level, the reviewer recommends a few revisions for further improvement.

We thank reviewer #2 for the kind comments and have addressed the concerns here and revised the manuscript in the result and discussion sections, highlighted in blue.

□

Major concerns:

#1) Figure 3f, j/page 5, line 193-195: please clarify the reason of why the Bassoon-Gephyrin pair does not form a synapse. Does the slightly larger gap (135 nm in j vs. 110-120 nm in h and i) provide a sufficient reason? Another concern is that a slight distortion during ExM could bias the gap distance. Is there any evidence to rule out that possibility?

We apologize for the confusion. The Bassoon-Gephyrin pair does indeed form a synapse. The point we were trying to make is that the cyan axon terminal runs close but does not provide the synapse onto the blue soma. This can be seen most clearly in **Fig. 3f**". Our intent was to show an example of avoiding synapse assignment mistakes. We added the following description to the figure legend to help clarify this point: "White arrow points to the Gephyrin-Bassoon pair that corresponds to an axo-somatic synapse between an unlabeled axon and the blue soma."

The reviewer makes a good point that although the expansion process is considered isotropic, slight distortions in expansion microscopy (1-5%, Tillber and Chen 2016 Nature Biotechnology) exist. On a 100 nm scale, this would be roughly equivalent to 1-5 nm error. Our gap distance measurements were performed not with the intention of precisely revealing the distance between Bassoon and Gephyrin as other studies have done, but with the intention of confirming that synapses could be adequately resolved and our expansion factor estimation gave us a measurement close to what had been previously reported. We would like to point out that spectral connectomics relies on adequately resolving synaptic pairs and assigning synapses between Brainbow neurons, a process that does not require precise measurements of distance.

#2) Mouse lines: In which experiment did the authors use Thy1-YFP-H strain? As long as the reviewer thoroughly read the manuscript, it might be used in the experiment in Supplementary Fig. 2. But there is no description regarding the mouse strain and the source of signals in neurons.

Thy1-YFP-H strain was indeed used in the experiment performed for Supplemental Figure 2. This information was included in Supplemental Table 1 and methods under "miriEx expansion measurement". To further clarify this point, additional description was added to Supplemental Figure 2.

#3) On page 3, line 96-97: SWITCH method (Murray et al. Cell 163, 1500–1514, 2015) intensively tested multiplex immunostaining with adopting a similar stripping-restaining strategy. The reviewer recommends to add it among the references.

We thank the reviewer for this suggestion and included SWITCH in the main text and as a reference.

Minor concerns

In Fig. 1b, information on which round of PV image was used in Merge panel should be clarified.

We clarified this point by adding additional description to Figure 1 legend: “The merged image shows r1 PV, r2 Calb, r3 Cb1R, r5 NOS, and r6 SERT.”

On page 3, line 121-127: please indicate the corresponding figure panel(s).

We added the corresponding figure panels (**Fig 1b**) to the main text.

On page 3, line 146: the figure reference (Fig. 2g-h) is apparently corresponding to Fig. 2j, k.

We thank the reviewer for pointing out this error. We corrected the figure reference to 2j-k to point readers to the morphology reconstructions.

Please provide more detailed descriptions in the legends of Supplementary Fig. 3 and 4.

We thank the reviewer for this point, and added the following additional descriptions in the legends of Fig S3 and S4:

Fig S3. “Example of how DAPI nuclear stain can be used as a fiduciary channel for registration. First, different rounds of DAPI stains are pre-aligned manually using Bigwarp plugin from Fiji/ImageJ. Then, the two rounds are automatically aligned using Elastix by using a non-linear B-spline grid. We found that a coarse pre-alignment step improved the Elastix registration. The resulting transformation was then applied to all the other antibody channels from round 1.”.

Fig S4. “100 µm formalin fixed human brain tissue was processed with miriEx to label 6 different targets across 2 rounds of immunostaining. DAPI was stained in each round to use as a fiduciary channel for registration.”

In Fig. 4e, Is the bar for n37 deliberately turned sideways?

We apologize for the confusion, the bar for n37 is not turned sideways. Instead, the number of unique axons and number of unique colors are the same value. To avoid confusion, we made the bar thinner in Fig. 4e.

Please thoroughly check typos in the manuscript, such as: “... doesn’t actually actually form a synapse...” (Fig. 3f legend); “brians were washed...” (Mouse perfusion and tissue sectioning in the Method section); AAx/Aas (miriEx protocol in the Methods section).

We thank the reviewer for pointing out these typos and we corrected them.

Reviewer #3 (Remarks to the Author):

This clearly written and beautifully illustrated manuscript describes several important individual innovations and combinations of same likely to be widely useful in many areas of neuroscience, including, but not limited to the specific focus area, "connectomics", called out in the present manuscript. I have no qualms about recommending publication in essentially the present form. I would suggest, however, that the writing should be clarified in two ways. (1) The term "dense" in the context of connectomic reconstruction is now coming to be understood in a fairly absolute sense to refer to reconstructing EVERY neuron. In this writing, it is used in a much more relative sense (i.e., to labeling and reconstructing at most all GABAergic neurons - a minor fraction of the total neuron population). The text should be modified to avoid this clash of meanings.

We thank the reviewer for this point. Indeed we used the term 'dense' in a relative sense compared to the previous standards of 'sparse' labeling in light microscopy-based neural reconstruction. We also agree with the reviewer that scientists have used the term "dense labeling/reconstruction" to mean "saturated labeling/reconstruction", especially in EM-based connectomics studies. We now changed all the terms of "dense labeling" to "dense labeling of a neuronal population" to specify the genetic labeling strategy, and removed "dense" when referring to reconstruction.

(2) A related caveat might also be in order: GABAergic neurons tend to have shorter and thicker axons and larger boutons, making them easier to identify spectrally and trace than the thinner and therefore optically "noisier" axons and boutons of the more populous glutamergic neurons. There should thus be a disclaimer that what is shown to be true here for GABA neurons might not be true for those generally much longer and skinnier glutamatergic axons and their often-tinier boutons. Having said that, the present report demonstrates methods of great promise, even if it did turn out to be as useful as claimed here only for GABAergic neurons. In any case, some discussion of this issue would certainly make this important contribution even more valuable.

The reviewer brings up a good point here. Different cell types have various axon morphology properties (thickness, bouton size, tortuosity, etc) that lead to varying levels of difficulty with reconstruction. To address this caveat, we added the following text to the discussion:

One important caveat to mention is that different neuronal cell types have various axon morphology properties that can be challenging to reconstruct. For example, glutamatergic cortical neurons can have smaller calibre axons, with tinier boutons, compared to the PV axonal arbors we reconstructed in this study. Although the concepts behind spectral connectomics can be applied broadly, Brainbow labeling (such as AAV titer and expression time) and imaging conditions (such as laser power and resolution) may need to be tailored for different subtypes of neurons.

REVIEWERS' COMMENTS:

Reviewer #1 (Remarks to the Author):

Ad #1: The authors explain the issue of resolving synapses in their rebuttal letter, yet they do not take any action apart from adding a movie showing the 3D situation, although the movie looks very convincing. I think it is necessary to point out in the manuscript why synapses are called putative and discuss the points the authors mentioned in their rebuttal letter.

Ad #4: There are several structures in Fig. S11B that are not traced. Why is that? The authors should mention in the legend for which reason some fluorescent structures are not traced. Also, a larger field of view with more dendrites and axons would be more informative than mostly showing a soma. The tutorial videos are convincing. Will these be referred to from the paper?

The authors mentioned that the estimated expansion factors were added to Table S1, however, I could not find them there. Also, another comment regarding Table S1 (number of confocal image planes) were not added to Table S1. Pls fix.

After resolving these remaining issues I enthusiastically support publication of this impressive study.

Reviewer #2 (Remarks to the Author):

The authors successfully revised all of my concerns.

REVIEWER COMMENTS

Reviewer #1 (Remarks to the Author):

Ad #1: The authors explain the issue of resolving synapses in their rebuttal letter, yet they do not take any action apart from adding a movie showing the 3D situation, although the movie looks very convincing. I think it is necessary to point out in the manuscript why synapses are called putative and discuss the points the authors mentioned in their rebuttal letter.

We edited the text in the revised manuscript, highlighted in blue on page 5, as suggested by the reviewer.

Ad #4: There are several structures in Fig. S11B that are not traced. Why is that? The authors should mention in the legend for which reason some fluorescent structures are not traced.

We added explanations in the legends of Fig. S11B to answer the none tracing question: "The goal of the tracing from this image is to analyze the inhibitory inputs to selected PV-neurons with their somas located within the imaging volume. To accelerate the tracing, we first identified the Bassoon-Gephyrin-Brainbow trio signals on the target PV-neurons followed by tracing all the input axons "outward". We ignored the axons passing by but not making connections to the target neurons, although they could be as confidently traced based on their similar labeling quality in the imaging."

Also, a larger field of view with more dendrites and axons would be more informative than mostly showing a soma.

The larger field of the overlay image is now attached as Fig. S11D.

The tutorial videos are convincing. Will these be referred to from the paper?

We added the link to the manual and tutorial videos where we cited nTracer in the main text (blue text on page 4) and in the methods (blue text on page 15).

The authors mentioned that the estimated expansion factors were added to Table S1, however, I could not find them there. Also, another comment regarding Table S1 (number of confocal image planes) were not added to Table S1. Pls fix.

Sorry that we provided the wrong table. The correct one with all fixes is attached as "Table S1.pdf".

Reviewer #2 (Remarks to the Author):

The authors successfully revised all of my concerns.